# Hydrodynamic pressure law of ground-rested circular RC tank under bi-directional horizontal seismic action

Lin Gao[1,2], Zengshun Chen[2], Huihui Yang[1], Xiaohui Wan[1], Mingzhen Wang[1]*

**1** School of Civil Engineering, Chongqing University of Arts and Sciences, Chongqing, China, **2** School of Civil Engineering, Chongqing University, Chongqing, China

* wmz917@126.com

**Data Availability Statement:** All relevant data are within the paper and its Supporting Information files.

## Abstract

The research object is the ground-rested circular RC tank. The innovation is to reveal the hydrodynamic pressure law of ground-rested circular RC tanks under bi-directional horizontal seismic action. The relationship between the sloshing wave height and hydrodynamic pressure is determined, the hydrodynamic pressure components and their combination are verified, calculation methods for hydrodynamic pressure are developed, and their distribution laws are presented. The results show that convective hydrodynamic pressure cannot be ignored when the tank is subjected to seismic action. Hydrodynamic pressure under uni-directional horizontal seismic action in X or Y direction is obtained by square root of the sum of impulsive pressure squared and convective pressure squared. Total hydrodynamic pressure under bi-directional horizontal seismic action is obtained by the square root of the sum of X-direction hydrodynamic pressure squared and Y-direction hydrodynamic pressure squared. This method can ensure the accuracy and reliability of hydrodynamic pressure calculation.

## 1 Introduction

Liquid storage structures widely exist in municipal engineering, petrochemical engineering, and nuclear engineering. Representative liquid storage structures include water storage tanks in water supply and drainage systems, oil storage tanks in the petrochemical industry, and liquid storage tanks in the nuclear industry [1]. These structures are functional structures and play an important role in the industry. However, the liquid storage structure is prone to structural damage and functional damage under previous strong earthquakes. And the resulting indirect loss is far greater than the direct loss [2]. Different from the analysis of the pier in the outer waters, the coupling between the structure and the inner waters belongs to the internal flow problem [3]. Structures and internal liquids exhibit different vibration characteristics when strong earthquakes occur. Liquid inertia and viscosity can dissipate part of the energy and play a certain energy dissipation effect, but at the same time, liquid sloshing will produce hydrodynamic pressure on the structure [4, 5]. Different from the ordinary building structure,

**Funding:** This study was supported by the Chongqing Natural Science Foundation of China (CSTC2019jcyj-msxmX0781) and the Youth Project of Science and Technology Research Program of Chongqing Education Commission of China (KJQN201901332). The funders provide financial support for this study.

**Competing interests:** The authors have declared that no competing interests exist.

the existence of liquid greatly improves the natural vibration period of the liquid-structure coupling system. The sloshing mechanism of liquid under long-period ground motion is different from that under short-period ground motion [6, 7]. The factors affecting the sloshing characteristics include objective factors such as site, epicentral distance, earthquake magnitude, and earthquake source characteristics, and subjective factors such as structure shape, size, and liquid storage height [8]. There are few studies on the distribution of hydrodynamic pressure for liquid storage structures under long-period ground motion action or bi-directional horizontal ground motion action [6, 9–11].

In order to avoid and reduce the direct, indirect, and secondary disasters caused by the damage of liquid storage structures in an earthquake, the seismic problem of liquid storage structures needs to be paid more attention. It is urgent to further study the liquid sloshing mechanism under the action of bi-directional horizontal ground motion with long period characteristics and establish a feasible and conservative calculation method of hydrodynamic pressure. The research results have important reference values for the safety and economical design of liquid storage structures.

## 2 Literature review

In 1957, Housner proposed the famous simplified calculation model of the hydrodynamic pressure for the rigid liquid-containing tank [12]. The hydrodynamic pressure effect of the internal flow for the liquid-containing tank was divided into the impulsive pressure component generated by the synchronous movement of the liquid with the tank and the convective pressure effect generated by the liquid sloshing for the first time. The Housner hydrodynamic pressure model is simple and practical and has high accuracy in calculating the liquid-structure coupling problem of large stiffness structures. It is still widely used in ACI 350.3–01 [13], API standard 650 (11th Edition) [14], and AWWA standard D100-96 [15]. Subsequently, many scholars studied the hydrodynamic pressure calculation and distribution law of flexible storage tanks. They believed that the hydrodynamic pressure calculation of flexible storage structure was more realistic than that of rigid assumption. In 1981, Haroun took the liquid in the tank as a continuum. According to the velocity potential theory and integral boundary method, the liquid mass was transformed into additional mass and applied to the tank wall. The calculation formulas of the liquid-structure coupling natural vibration period and the maximum sloshing wave height were given, which were used to calculate the dynamic characteristics and vibration response of the liquid-containing structure under seismic action. The theoretical analysis results were verified by relevant tests [16]. Hwang and Ting used the boundary element method to divide the liquid and the tank into two parts. The two parts were linked by hydrodynamic pressure, and finally, hydrodynamic pressure is combined into the finite element equation of the tank [17]. Yu and Whittaker studied the vibration frequency, hydrodynamic pressure, sloshing wave height, and foundation response of the liquid-containing structure by using the FSI analysis method. It is considered that ignoring the flexibility of the liquid-containing structure will significantly underestimate the seismic response of the structure and the liquid contained therein [18].

Ghaemmaghami and Kianoush used the finite element method to study the seismic performance of the two-dimensional flexible rectangular liquid-containing structure in 2010 [19]. It is found that the impulse response and convection response of the liquid-structure coupling system under vertical seismic action to the structure are far less than that under horizontal ground motion. Kianoush and Ghaemmaghami used the finite element method to simulate the soil-structure-liquid interaction [20] numerically. It was found that the seismic response of the soil-structure-liquid coupling system was very sensitive to the spectral characteristics of the

seismic input by analyzing different ground motion input results. A simple model with viscous boundary can be used to the fundamental variability effect, including a linear elastic medium. At the same time, the responses of the 'fat and short' and 'slim and tall 'flexible liquid-containing structures under horizontal and vertical seismic actions are also analyzed. Moslemi and Kianoush researched the influence of liquid free surface sloshing, tank stiffness, tank height-width ratio, vertical seismic action, and tank bottom constraint on the dynamic response of the tank [8]. The seismic design suggestions for cylindrical liquid-containing tanks were put forward according to the calculation results, and the design of hydrodynamic pressure in the ACI 350.3–01 code [13] was too conservative. Hashemi et al. proposed a method to calculate the dynamic response of the three-dimensional rectangular liquid-containing structure of the flexible tank [21]. The influence of liquid-structure interaction on the dynamic response of the liquid-containing structure with partial storage capacity was considered when evaluating the impulsive pressure. The velocity potential energy satisfying the boundary conditions was calculated by using the superposition principle to separate variables. A program was developed to calculate the convective pressure and liquid surface sloshing displacement. Park et al. carried out the dynamic test of the cylindrical liquid-containing tank under horizontal seismic action and determined the occurrence of the beam and elliptical vibration modes according to the acceleration results measured at different positions [22]. Kotrasova and Kormanikova investigated the influence of liquid on the tank when the liquid-containing structure was subjected to horizontal acceleration [23]. Hydrodynamic pressure, including impulsive pressure and convective pressure, was excited, and the seismic responses of multiple sizes of circular liquid-containing structures fixed on a rigid basis were calculated. The calculation program has been adopted by Eurocode 8. ADINA finite element software was used to analyze the dynamic characteristics, sloshing wave height, and hydrodynamic pressure of vertical cylindrical oil storage tank in reference [24]. It was found that the tank radius and the liquid-containing height had a great influence on the hydrodynamic pressure, and the ratio of hydrodynamic pressure to hydrostatic pressure was related to the liquid-containing height. The calculation results of hydrodynamic pressure in the Chinese Code for the design of vertical cylindrical welded steel oil tanks (GB 50341–2014) [25] were reliable. Pandit et al. measured the impulsive and convective responses in the hydrodynamic pressure of the liquid-containing tank with the inclined wall through multi-condition numerical simulation and self-compiling MATLAB program method [26]. So far, the flexible wall analysis methods considering liquid sloshing coupling include the assumed mode method, Haroun-Housner method, and finite element method.

## 3 Methodology

### 3.1 Calculation method of wave height for liquid sloshing

The maximum sloshing wave height of the liquid surface for the water tank represented by $h$ is calculated by Eq (1) [27]. In Eq (1), the parameter $R$ is the tank inner radius. The parameter $\beta_1$ is directly obtained based on the long-period response spectrum in Fig 1 [28] according to the first-order sloshing period $T_1$. The parameter $k$ is the horizontal seismic coefficient corresponding to the fortification intensity, for which 0.1 is taken at 7˚, 0.2 is taken at 8˚, and 0.4 is taken at 9˚.

$$h = 3.662 R \beta_1 k \tag{1}$$

$\beta_{max}$ in Fig 1 is equal to 2.25. $T_g$ in Fig 1 is site characteristic period and its values are detailed in Table 1.

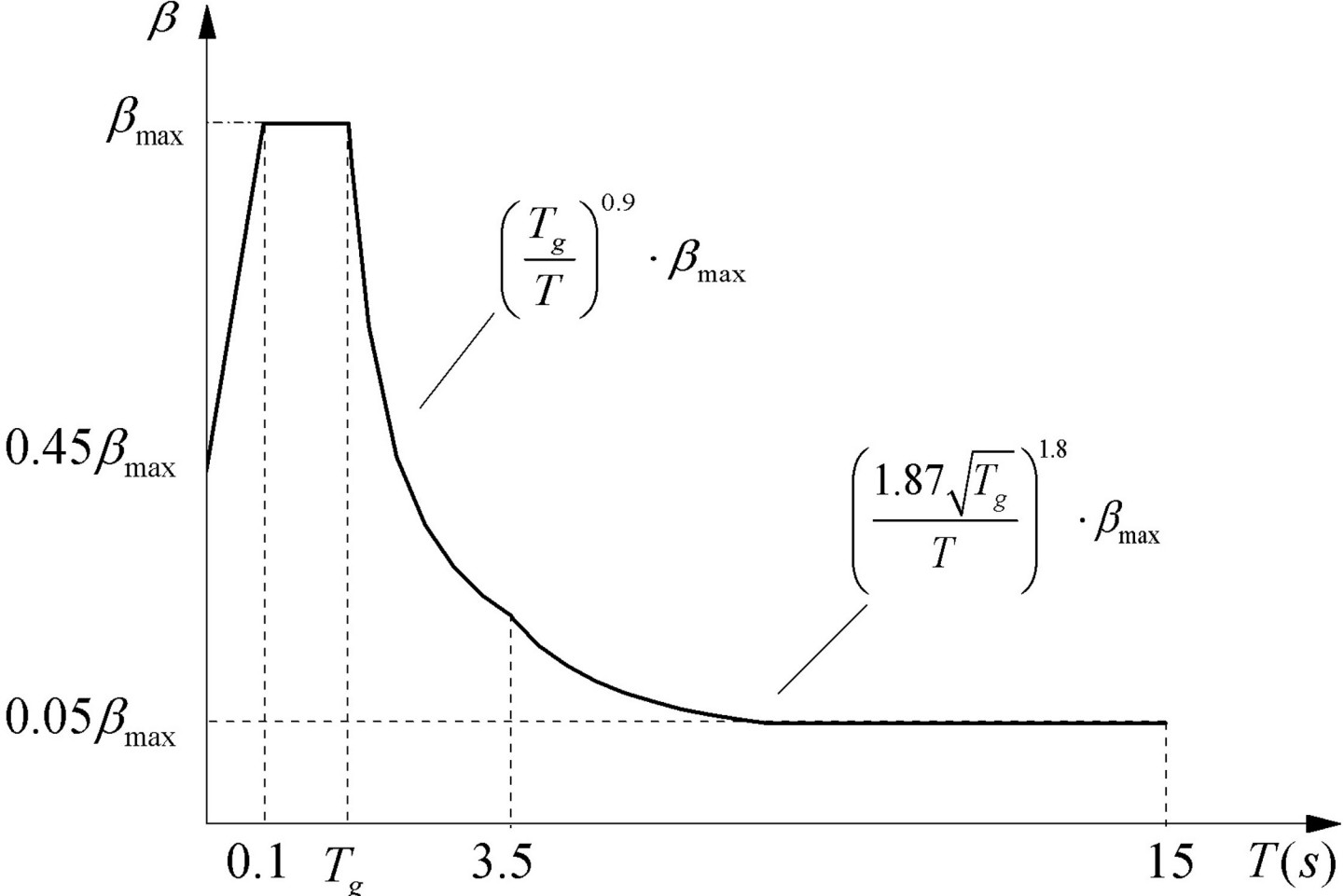

**Fig 1. Long-period seismic design $\beta$ spectrum with 5% damping ratio.**

## 3.2 Theoretical calculation method of hydrodynamic pressure

The equation of calculating the total pressure at any point in the liquid is [29, 30]:

$$p(r, \theta, z, t) = \rho \left( \frac{\partial \phi}{\partial t} + gz \right) \tag{2}$$

Where: the parameter $r$ is the vertical distance from the calculation point to the centerline of the liquid storage structure. $\theta$ is the angle in the circumferential direction. $z$ is the depth of the liquid. $t$ is the time. $\phi$ is velocity potential. And $g$ is the acceleration of gravity. The meaning of the above parameters can be understood by combining Fig 2.

**Table 1. The values of site characteristic period $T_g$.**

| Seismic Design Group | Site Classification | | | | |
|---|---|---|---|---|---|
| | $I_0$ | $I_1$ | II | III | IV |
| First group | 0.20 | 0.25 | 0.35 | 0.45 | 0.65 |
| Second Group | 0.25 | 0.30 | 0.40 | 0.55 | 0.75 |
| Third Group | 0.30 | 0.35 | 0.45 | 0.65 | 0.90 |

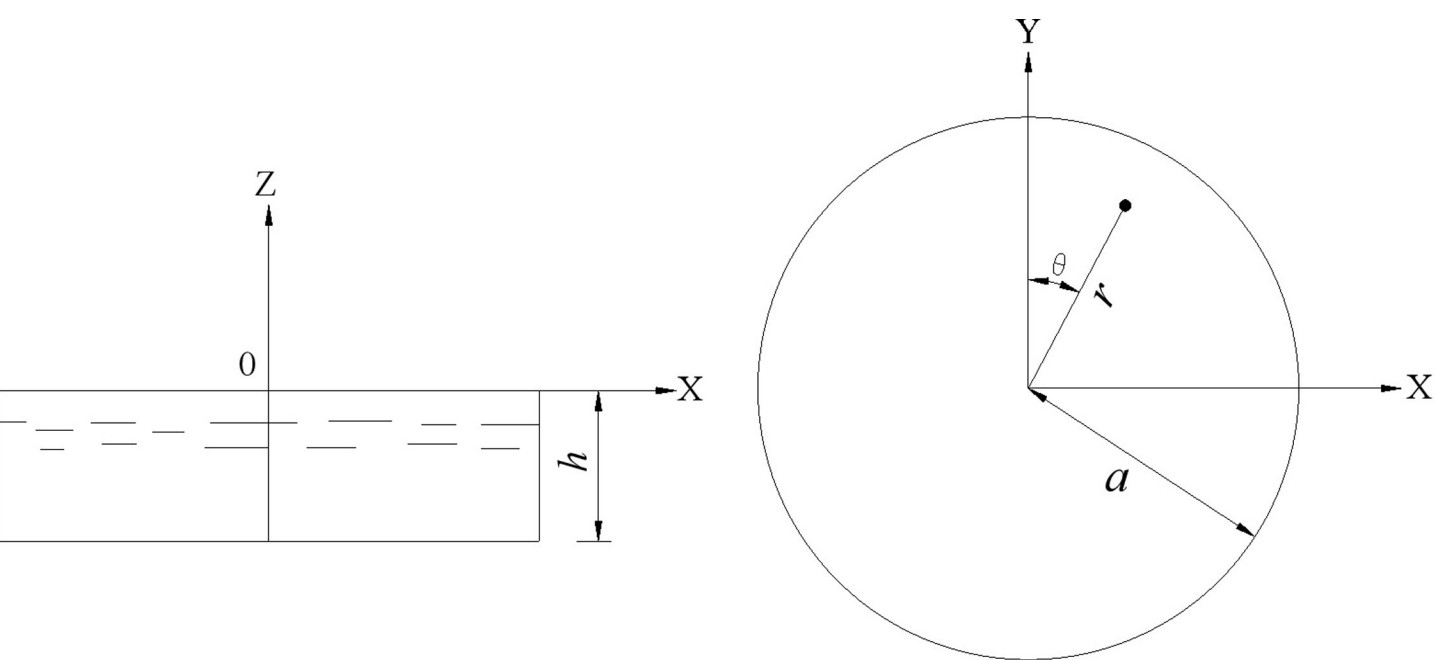

**Fig 2. Schematic diagram of geometry and coordinate system of the circular tank.**

Eq (2) shows that the total pressure $p(r, \theta, z, t)$ of the liquid is the sum of the hydrodynamic pressure and the hydrostatic pressure. The calculation formula of hydrodynamic pressure is $p(r, \theta, z, t) = \rho \frac{\partial \phi}{\partial t}$, and the calculation formula of hydrostatic pressure is $p(r, \theta, z) = \rho g z$.

In the theoretical analysis of hydrodynamic pressure, it is assumed that the tank structure is a rigid body and contains ideal fluid (no viscosity, no rotation, and incompressible). The velocity potential $\phi$ of the liquid satisfies the Laplace equation:

$$\nabla^2 \phi = 0 \tag{3}$$

The corresponding hydrodynamic pressure is:

$$p = -\rho \frac{\partial \phi}{\partial t} \tag{4}$$

Where, $\rho$ is the density of the liquid.

The boundary conditions in theoretical research are as follows:

① The movement velocity of the wall for the liquid storage structure should be consistent with that of the ground named $\dot{x}_0(t)$.

② The normal velocity at the bottom is equal to zero. That is $\frac{\partial \phi}{\partial n} = 0$. $n$ is the unit vector in the normal direction.

③ The equilibrium condition of the liquid surface under the influence of gravity field is

$$u = -\frac{1}{g} \frac{\partial \phi}{\partial t} \tag{5}$$

Where $u$ is the wave height of the liquid surface and $g$ is the acceleration of gravity.

Fig 2 shows the geometry and coordinate system of the circular tank. Under the above assumptions, the velocity potential $\phi$ of the liquid and the hydrodynamic pressure on the tank wall can be solved by the separation variable method.

1) Hydrodynamic pressure acting on tank wall

$$p^w(z,t) = -A\left[\ddot{x}_0(t) + \sum_{j=1}^{\infty} d_j p_j^w(z)\ddot{q}_j(t)\right] \tag{6}$$

In the Eq (6), the undamped generalized coordinates $\ddot{q}_j(t)$ are determined according to the differential equation of Eq (7), that is

$$\ddot{q}_j(t) + \tilde{\omega}_j^2 q_j(t) = -\ddot{x}_0(t) \tag{7}$$

For the damped viscous fluid, it is approximately determined according to Eq (8), that is

$$\ddot{q}_j(t) + 2\varepsilon_j\tilde{\omega}_j\dot{q}_j(t) + \tilde{\omega}_j^2 q_j(t) = -\ddot{x}_0(t) \tag{8}$$

Where, $\ddot{x}_0(t)$ is the acceleration component of horizontal ground motion; $\varepsilon_j$ is the critical damping ratio of the $j$th mode shape of the liquid. In order to simplify the calculation, the concept of mode damping is adopted. When solving the differential of Eqs (7) and (8), the initial conditions given by the velocity potential should be satisfied.

In Eqs (6) to (8), the coefficients are calculated as follows:

$$\begin{cases} p_j^w(z) = \dfrac{\cosh(\sigma_j z/a)}{\cosh(\sigma_j h/a)} \\ \tilde{\omega}_j^2 = \dfrac{g}{a}\sigma_j \tanh(\sigma_j h/a) \end{cases} \tag{9}$$

For a circular tank,

$$\begin{cases} A = \rho a \sin\theta \\ d_j = \dfrac{2}{\sigma_j^2 - 1} \end{cases} \tag{10}$$

$\sigma_j$ is the $j$th root of $J_1'(\sigma_n)$ by the derivative of the first-order Bessel function $J_1(\sigma_n)$, and the values of the first six roots are 1.84, 5.33, 8.53, 11.71, 14.86, and 18.00 in sequence. The corresponding $d_j$ values are 0.837, 0.073, 0.0279, 0.0147, 0.0091 and 0.0062 in sequence. The parameter $\rho$ is the density of the liquid. $a$ is the inner radius of the liquid storage structure. $\theta$ is the angle in the circumferential direction.

There are the following relationships for coefficients $d_j$:

$$\sum_{j=1}^{\infty} d_j = 1 \tag{11}$$

For a circular tank, the symbol $\sum_{j=1}^{\infty}$ means the sum of $j$ from 1, 2, 3, and so on in turn.

2) Hydrodynamic pressure acting on the tank bottom

The hydrodynamic pressure of the tank bottom for the circular tank can also be determined according to Eq (4) of velocity potential, and the calculation equation is as follows.

$$p_c^b(r,t) = -A\left[\frac{r}{a}\ddot{x}_0(t) + \sum_{j=1}^{\infty} d_j \ddot{q}_j(t) R_j\left(\frac{\sigma_j r}{a}\right)\right] \tag{12}$$

Where, $R_j\left(\frac{\sigma_j r}{a}\right) = \frac{1}{\cosh(\sigma_j h/a)} \cdot \frac{J_1\left(\frac{\sigma_j r}{a}\right)}{J_1(\sigma_j)}$. $R_j\left(\frac{\sigma_j r}{a}\right)$ is the $j$th mode shape function.

$A$ and $d_j$ are the same as Eq (10), $J_1(\sigma_n)$ is a first-order Bessel function, and the values of $\sigma_j$ are as mentioned before.

It can be seen from Eqs (6) and (12) that the hydrodynamic pressure acting on the wall and bottom of the tank structure is similar to the seismic response of the elastic structure in form. The coefficient $d_j$ is equivalent to the mode participation coefficient of the elastic structure. $p_j^w(z)$ and $R_j(\sigma_j \frac{r}{a})$ are equivalent to the shape mode functions. $\ddot{q}_j(t)$ is the generalized coordinate. So the Eqs (6) and (12) have the same meanings as the general elastic structure. The response spectrum theory can be used to calculate the seismic response of hydrodynamic pressure. The design of the engineering structure is not concerned about the real-time value of hydrodynamic pressure and structural response at every moment under every earthquake process, but the most concerned is the maximum hydrodynamic pressure and structure response in the tank structure under a certain earthquake. The hydrodynamic pressure can be solved by response spectrum theory.

The Eqs (6) and (12) are rewritten as follows when the response spectrum theory is used to solve the hydrodynamic pressure:

$$p^w(z,t) = -A\left[\left(1 - \sum_{j=1}^{\infty} d_j p_j^w(z)\right)\ddot{x}_0(t) + \sum_{j=1}^{\infty} d_j p_j^w(z)\left(\ddot{x}_0(t) + \ddot{q}_j(t)\right)\right] \tag{13}$$

$$p_c^b(r,t) = -A\left\{\left[\frac{r}{a} - \sum_{j=1}^{\infty} d_j R_j\left(\frac{\sigma_j r}{a}\right)\right]\ddot{x}_0(t) + \sum_{j=1}^{\infty} d_j R_j\left(\frac{\sigma_j r}{a}\right)\left(\ddot{x}_0(t) + \ddot{q}_j(t)\right)\right\} \tag{14}$$

The first item in the square brackets of Eqs (13) and (14) is the impulsive pressure, which is the hydrodynamic pressure when the ground motion is close to the instantaneous pulse. Since the interference frequency is much larger than the natural frequency of the liquid, so the liquid level does not vibrate. The impulsive pressure is the hydrodynamic pressure when the boundary condition of the liquid surface is $u = -\frac{1}{g}\frac{\partial \varphi}{\partial t} = 0$ or $\varphi = 0$. The time-variant changing process of the impulsive pressure is entirely consistent with the ground acceleration $\ddot{x}_0(t)$, and it has positive correlation relation. The proportional coefficient is $A\left(1 - \sum_{j=1}^{\infty} d_j p_j^w(z)\right)$ and

$A\left[\frac{r}{a} - \sum_{j=1}^{\infty} d_j R_j\left(\frac{\sigma_j r}{a}\right)\right]$, which can be regarded as the equivalent mass distributed with the depth and the size of the tank bottom.

The second term in the square brackets of Eqs (13) and (14) is the convective pressure, which is formed by the product of infinitely many distributed masses named $A d_j p_j^w(z)$ or $A d_j R_j\left(\frac{\sigma_j r}{a}\right)$ and time functions $\left(\ddot{x}_0(t) + \ddot{q}_j(t)\right)$, corresponding to the hydrodynamic pressure of the $j$th mode of infinitely many liquid modes. Each time function $\left(\ddot{x}_0(t) + \ddot{q}_j(t)\right)$ in Eqs (13) and (14) represent the absolute acceleration response of a single-mass system with frequency

$\tilde{\omega}_j$ and mode damping coefficient $\zeta_j$ under the action of seismic ground acceleration $\ddot{x}_0(t)$, and its maximum value is equal to $gk\beta_j$. Where, $k$ is the seismic coefficient, which is the maximum acceleration of the ground in terms of the acceleration of gravity $g$, $k = (\ddot{x}|_{max}/g)$. And $\beta_j$ is the dynamic amplification coefficient of the $j$th mode, $\beta_j = \left[\ddot{x}_0 + \ddot{q}_j|_{max}/(\ddot{x}_0|_{max})\right]$.

In Eqs (13) and (14), their maximum values generally do not occur at the same time because $\ddot{x}_0(t)$ and $\left(\ddot{x}_0(t) + \ddot{q}_j(t)\right)$ ($j = 1, 2, 3\cdots$) are different time functions, and it is too conservative to superimpose the maximum values of the two kinds of pressures at the same time. The square root of the sum of squares (SRSS) method is generally applied to the combination of internal forces. That is to say, the internal forces, including impulsive pressure and convection pressure generated from each mode, should be calculated respectively, and then the SRSS method is used to combine these internal forces to get the maximum internal force. However, the calculation is complicated. Considering that the hydrodynamic pressure does not change the negative or positive sign repeatedly like the seismic force of each mode for multi-point elastic structure, the error caused by the direct combined pressure is small. Therefore, the SRSS method is used to calculate the maximum hydrodynamic pressure on the tank structure in the theoretical calculation.

1) The maximum hydrodynamic pressure acting on the wall of the circular tank is:

$$p_{max}^w(z) = Akg\left\{\left[1 - \sum_{j=1}^{\infty} d_j p_j^w(z)\right]^2 + \sum_{j=1}^{\infty}\left[\beta_j d_j p_j^w(z)\right]^2\right\}^{\frac{1}{2}} \tag{15}$$

2) The maximum hydrodynamic pressure acting on the bottom plate of the circular tank is:

$$p_{c,max}^b(r) = Akg\left\{\left[\frac{r}{a} - \sum_{j=1}^{\infty} d_j R_j\left(\frac{\sigma_j r}{a}\right)\right]^2 + \sum_{j=1}^{\infty}\left[\beta_j d_j R_j\left(\frac{\sigma_j r}{a}\right)\right]^2\right\}^{\frac{1}{2}} \tag{16}$$

### 3.3 Code calculation method of hydrodynamic pressure

A hydrodynamic pressure distribution map and the standard calculation formulas for the wall and bottom plate of a circular water tank are given in provision 6.2.2 of "Code for seismic design of outdoor water supply, sewerage, gas, and heating Engineering (GB 50032–2003) [31]", which are shown in Fig 3 and Eq (17), respectively.

$$F_{wc,k}(\theta) = K_H \cdot \gamma_W \cdot H_W \cdot f_{wc} \cos\theta \tag{17}$$

where $F_{wc,k}(\theta)$—the standard hydrodynamic pressure value for a circular tank at $\theta$ angle (kN/m$^2$),

$K_H$—seismic coefficient, that is, the ratio of the horizontal seismic acceleration to the gravitational acceleration, for which 0.05 is taken at 6 degrees, 0.1 is taken at 7 degrees, 0.15 is taken at 7 degrees, 0.2 is taken at 8 degrees, 0.3 is taken at 8 degrees, and 0.4 is taken at 9 degrees.

$\gamma_W$—the bulk density of water (kN/m$^3$).

$H_W$—water storage height (m).

$\theta$—the angle between the calculated section and the axis of the seismic input direction.

$f_{wc}$—the hydrodynamic pressure coefficient of a circular tank, adopted according to Table 2.

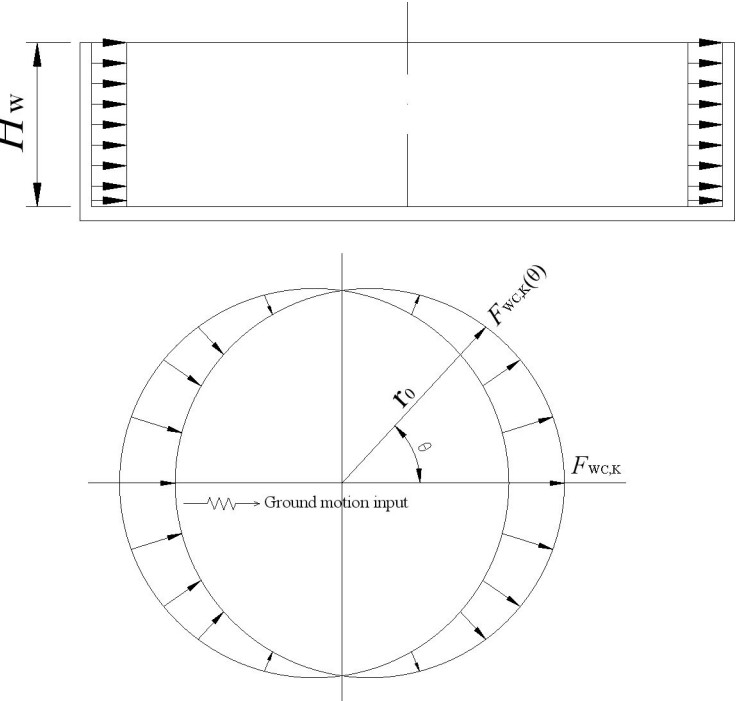

**Fig 3. Hydrodynamic pressure distribution of a circular tank in GB 50032–2003 code.** a) Along the height of the tank wall. b) Along the circumferential direction of the tank bottom.

It can be seen from Fig 3 and Eq (17) that the hydrodynamic pressure is at a maximum at the circumferential point along the seismic input direction, the hydrodynamic pressure is zero at the circumferential point vertical to the seismic input direction, and the hydrodynamic pressure is distributed along the circumferential direction as the *cos* function. The hydrodynamic pressure is distributed in a rectangular shape along with the height of the tank wall. The magnitude of hydrodynamic pressure is related to the following five factors: seismic coefficient, liquid bulk density, water storage height, tank radius, and calculation point location.

The distribution of hydrodynamic pressure on the tank wall of the circular tank is given in Fig R5.2 of Chapter 5 for ACI 350.3–01 code [13], as shown in Fig 4. The distribution pattern of hydrodynamic pressure is the same as in Fig 3B) from GB 50032–2003 Chinese code [31]. The hydrodynamic pressure formed by 1/2 impulsive force plus 1/2 convective force is specially indicated in Fig 4. Because the distribution form of hydrodynamic pressure in the two codes is the same, the calculation results of the GB 50032–2003 Chinese code are mainly used in the following comparison.

## 3.4 Numerical simulation calculation method of hydrodynamic pressure

The types of the analyzed tanks are ground-rested circular reinforced concrete with capacities of 500m³, 200m³, and 2000m³, hereinafter referred to as tanks A, B, and C, respectively. Table 3 lists the structure characteristics of the analyzed tanks. Table 4 lists the corresponding

**Table 2. Hydrodynamic pressure coefficient $f_{wc}$ of a circular tank.**

| $H_W/R$ | ≤0.6 | 0.8 | 1.0 | 1.2 | 1.4 | 1.6 | 1.8 | 2.0 | 2.2 |
|---|---|---|---|---|---|---|---|---|---|
| $f_{wc}$ | 0.40 | 0.39 | 0.36 | 0.34 | 0.32 | 0.30 | 0.28 | 0.26 | 0.25 |

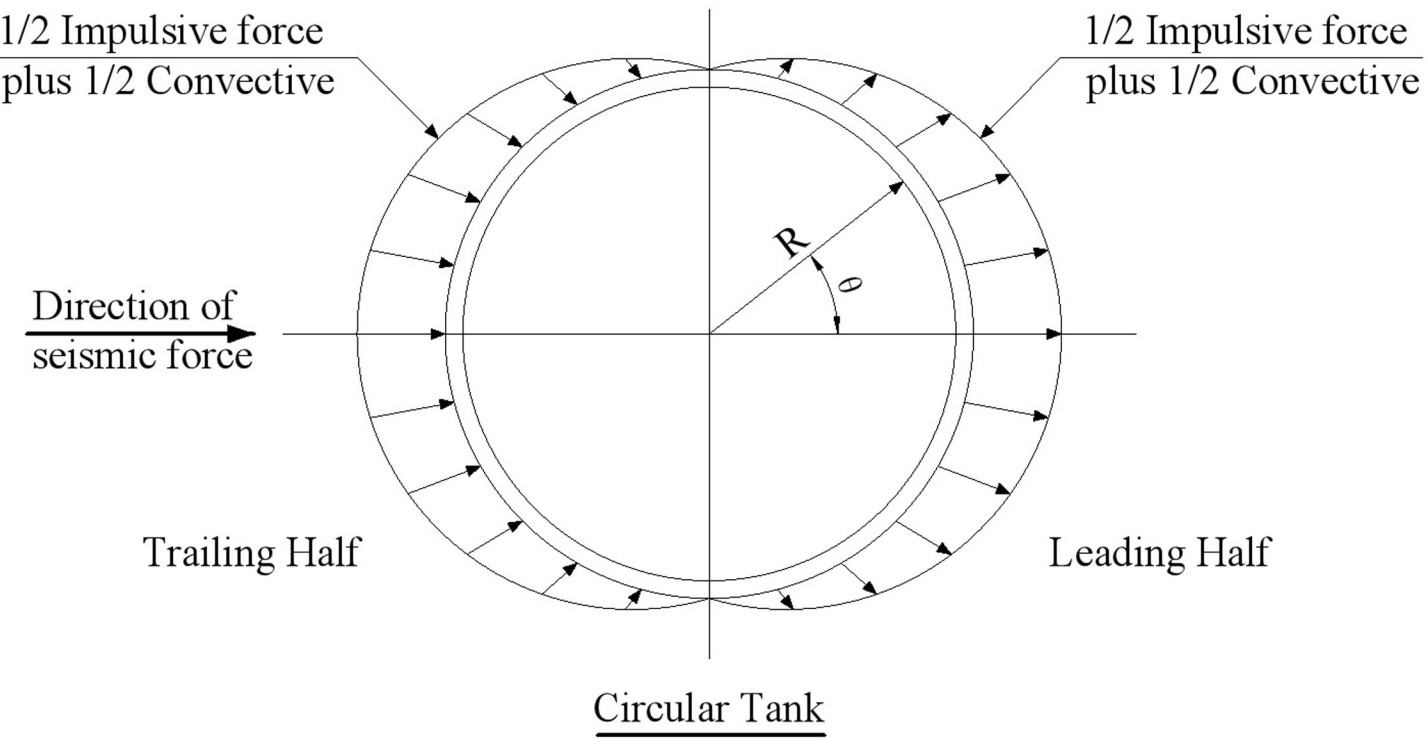

**Fig 4. Hydrodynamic pressure distribution of a circular tank in ACI 350.3–01 code.**

relationship between water storage capacity and water storage height. Table 5 lists the material properties of liquid water.

In order to simplify the calculation condition of the tank, the symbol "A-50%" is used to represent that the 500m$^3$ capacity tank has 1.75 meters water storage height. In ADINA software, ADINA Parasolid geometric modeling method is used to establish the tank model. The tank structure is adopted the 3D-Solid element. The concrete material is simulated by Concrete in ADINA, and the reinforcement is set by the Rebar option in the Truss element. The liquid in the tank is adopted the 3D-Fuild element. Thereinto, liner potential-based element is used in static analysis and modal analysis, and potential-based fluid is used in dynamic analysis. The stress-strain curves of the concrete and reinforcement are shown in Fig 5. The bottom of the tank structure is the fixed constraint. In order to make the grid division uniform and improve computational efficiency, the following method is used to divide the grid. The tank body in the direction of tank wall thickness and the bottom plate thickness is divided into three parts, and the tank body in the circumferential direction is divided into 50 parts. Each 0.35 meters along the tank wall height direction is divided into one portion. The radial direction of tanks A, B, and C are divided into 23, 15, and 37 parts, respectively. The circumferential and radial grids of the liquid in the tank are the same as those in the tank body, and the liquid

**Table 3. Structure characteristics of the circular tank.**

| Shortname | Capacity(m³) | Bottom thickness(m) | Wall thickness(m) | Inner radius(m) | The maximum water storage height(m) | Reinforcement diameter |
|---|---|---|---|---|---|---|
| A | 500 | 0.3 | 0.25 | 6.75 | 3.5 | 10mm |
| B | 200 | 0.3 | 0.25 | 4.3 | 3.5 | 10mm |
| C | 2000 | 0.3 | 0.25 | 13.5 | 3.5 | 10mm |

**Table 4. The Corresponding relationship between water storage capacity and water storage height.**

| Water storage capacity | No water | 10% | 20% | 30% | 40% | 50% | 60% | 70% |
|---|---|---|---|---|---|---|---|---|
| Water storage height (m) | 0 | 0.35 | 0.70 | 1.05 | 1.40 | 1.75 | 2.10 | 2.45 |

is divided one portion per 0.35 meters in the height direction. Taking tank A as an example, the finite element models of the tank body, reinforcement bar, 30% water storage, and 70% water storage are listed in Fig 6.

### 3.5 Seismic ground motion selection and input method

This research is to explore the seismic response characteristics of storage structure under long-period ground motion, so the obvious difference in spectrum characteristics, including extremely short period, short period, medium-long period, and long period are selected. At the same time, seven natural ground motions in the 2008 Wenchuan Earthquake in China with the peak acceleration of $1m/s^2$ are used as input for analyses [32]. Considering the accuracy and efficiency of calculation, the duration time of ground motion is taken as 30 seconds. The following information of seismic ground motions is collected and listed in Table 6, including name, collecting stations, site condition, epicentral distance, direction, peak acceleration, the moment of peak acceleration, and predominant period. At the same time, two groups of classical ground motions, including El Centro and Tianjin ground motion, are selected as ground motion inputs. The predominant period is a key parameter for the seismic design of important structures. The surface soil layer has a selective amplification effect on seismic waves of different periods, resulting in the waveform of some periods on the seismic record map being particularly many and good, which is called 'predominant', so it is called the predominant period of ground motion. The predominant period is the period where the maximum amplitude of soil vibration may occur, which mainly changes with the geotechnical characteristics of the site. When carrying out the seismic design of the structure, the natural vibration period of the structure and the predominant period of different foundations should be considered to ensure that the natural vibration period of the structure is greatly different from the predominant period of the site [33]. The predominant period of ground motion can be obtained through Fourier transform [34]. The selected ground motions, including long-period and short-period divided by the predominant period, are used to analyze the effects of ground motion characteristics on hydrodynamic pressure. In each group of ground motions, the ground motion in the direction with larger peak acceleration is defined as the main ground motion, including HSDB-EW, JZGYF-NS, PJW-NS, BJ-EW, CC-NS, YL-NS, and HX-NS, which is input to the X direction of the tank models in the calculation under unidirectional seismic action input. The remaining ground motion in the perpendicular direction of each group ground motion is called secondary ground motion, which is input to the Y direction of the tank model and simultaneously input with the main ground motion in the bi-directional ground motion calculation. Fig 7 shows the time histories of the ground motions. The difference in the predominant periods of the two group ground motions is directly reflected in the apparent difference in the shape of the time-history curve. Gravity acceleration $g$ is taken as $10m/s^2$.

**Table 5. Material properties of liquid water.**

| Density (Kg/m$^3$) | Bulk modulus (Pa) | Damping ratio |
|---|---|---|
| 1000 | $2.3 \times 10^9$ | 0.16% |

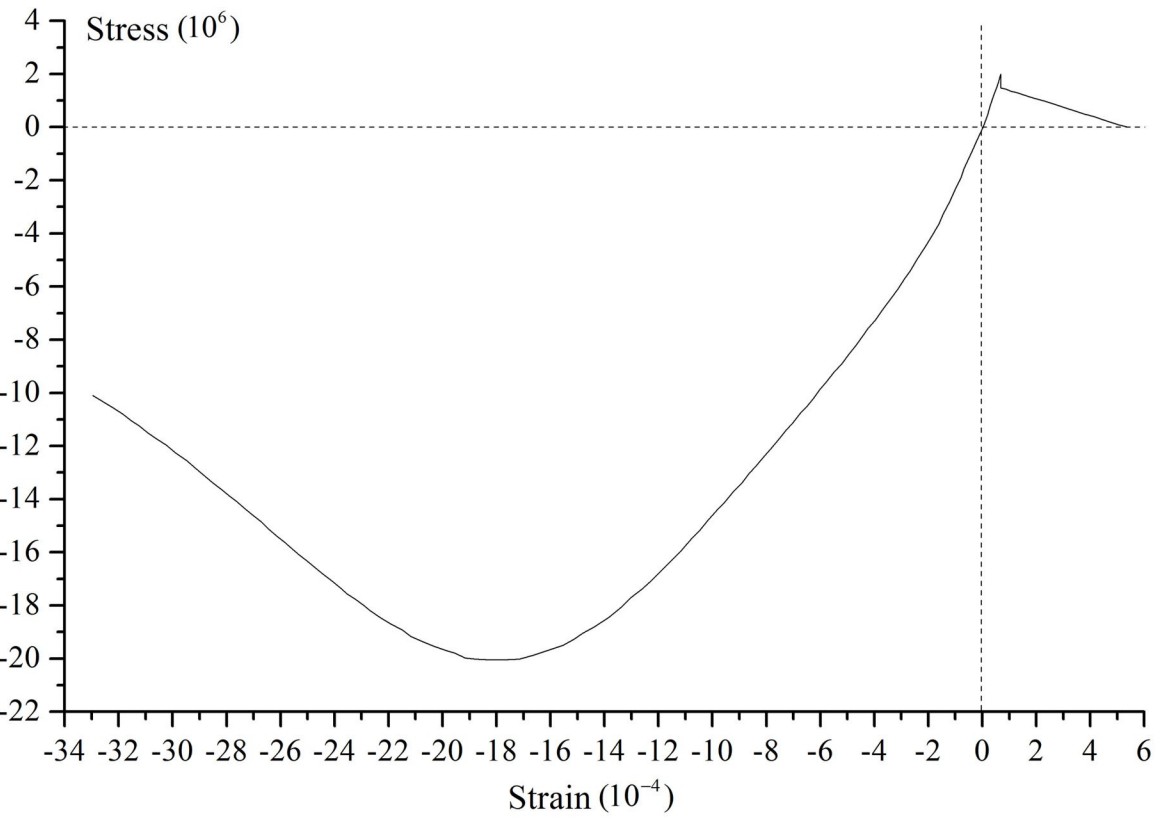

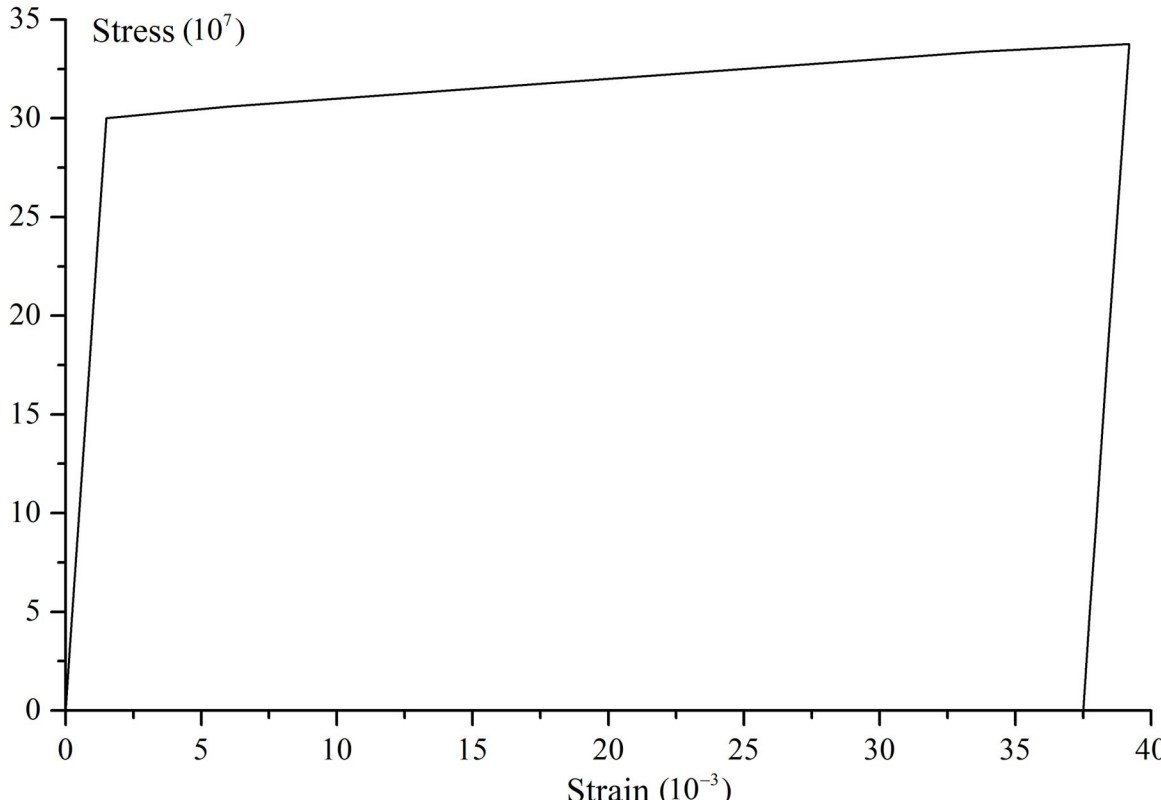

**Fig 5. The stress-strain curve of the materials in the numerical simulation.** a) The concrete. b) The reinforcement.

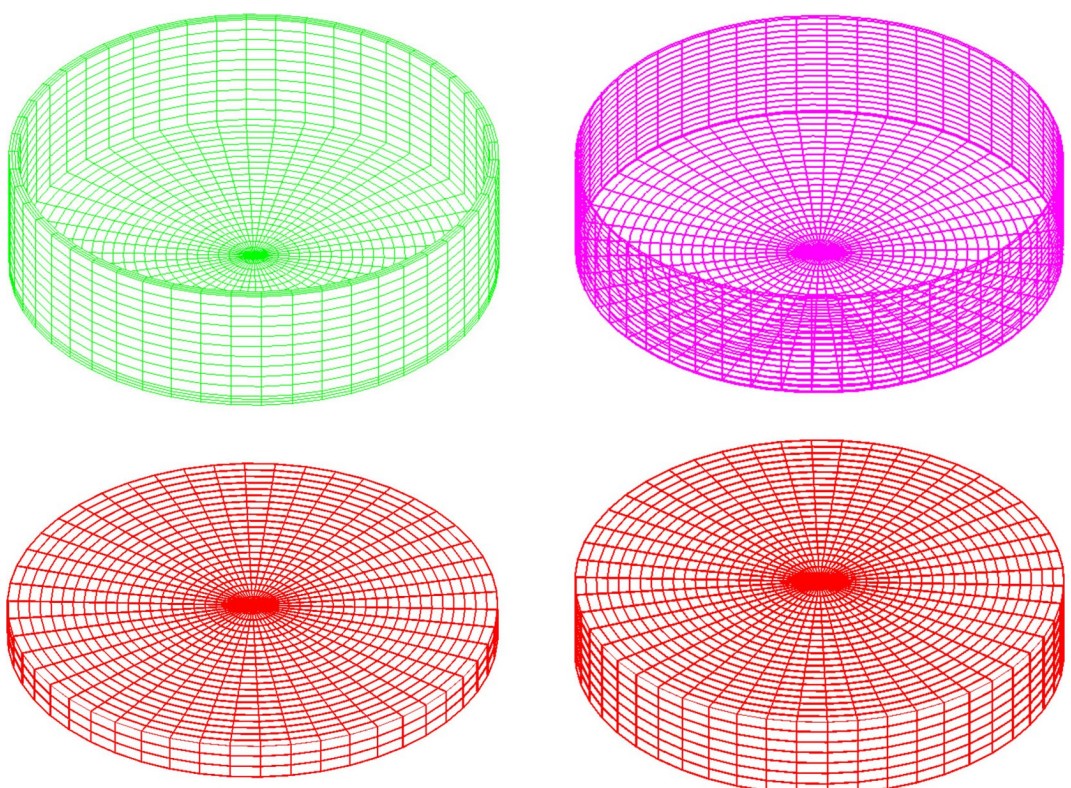

**Fig 6. The finite element models of tank A.** a) Tank body. b) Reinforcement bar. c) 30% water storage. d) 70% water storage.

Table 6. Basic information of seismic ground motions.

| Name, collecting stations | Site condition | The epicentral distance (km) | Direction | Peak acceleration ($10^{-2}$ m/s$^2$) | Moment of peak acceleration (s) | Predominant period (s) |
|---|---|---|---|---|---|---|
| HSDB Sichuan Province | II class | 125.9 | EW (main) | -102.643 | 6.160 | 0.095 |
| | | | NS (secondary) | 89.129 | 2.405 | 0.111 |
| JZGYF Sichuan Province | II class | 263.3 | EW (secondary) | -91.813 | 6.745 | 0.151 |
| | | | NS (main) | 100.224 | 18.425 | 0.168 |
| PJW Sichuan Province | II class | 81.0 | EW (secondary) | 97.454 | 7.405 | 0.310 |
| | | | NS (main) | 101.149 | 10.105 | 0.297 |
| BJ Shanxi Province | III class | 513.1 | EW (main) | 120.292 | 15.720 | 0.611 |
| | | | NS (secondary) | -90.121 | 16.430 | 0.532 |
| CC Shanxi Province | III class | 528.6 | EW (secondary) | 91.260 | 10.235 | 0.719 |
| | | | NS (main) | 107.719 | 14.980 | 1.138 |
| YL Shanxi Province | III class | 569.9 | EW (secondary) | 77.602 | 15.750 | 1.950 |
| | | | NS (main) | -94.005 | 16.730 | 1.862 |
| HX Shanxi Province | III class | 599.0 | EW (secondary) | 67.806 | 24.885 | 1.138 |
| | | | NS (main) | 92.090 | 13.835 | 4.096 |
| El Centro in Imperial Valley Eaerhquake | II class | 12.99 | EW (secondary) | 210.100 | 11.440 | 0.468 |
| | | | NS (main) | 341.700 | 2.120 | 0.683 |
| Tian Jin in Ninghe Earthquake | III class | 65 | EW (secondary) | 104.18 | 7.58 | 1.365 |
| | | | NS (main) | 145.80 | 7.64 | 0.890 |

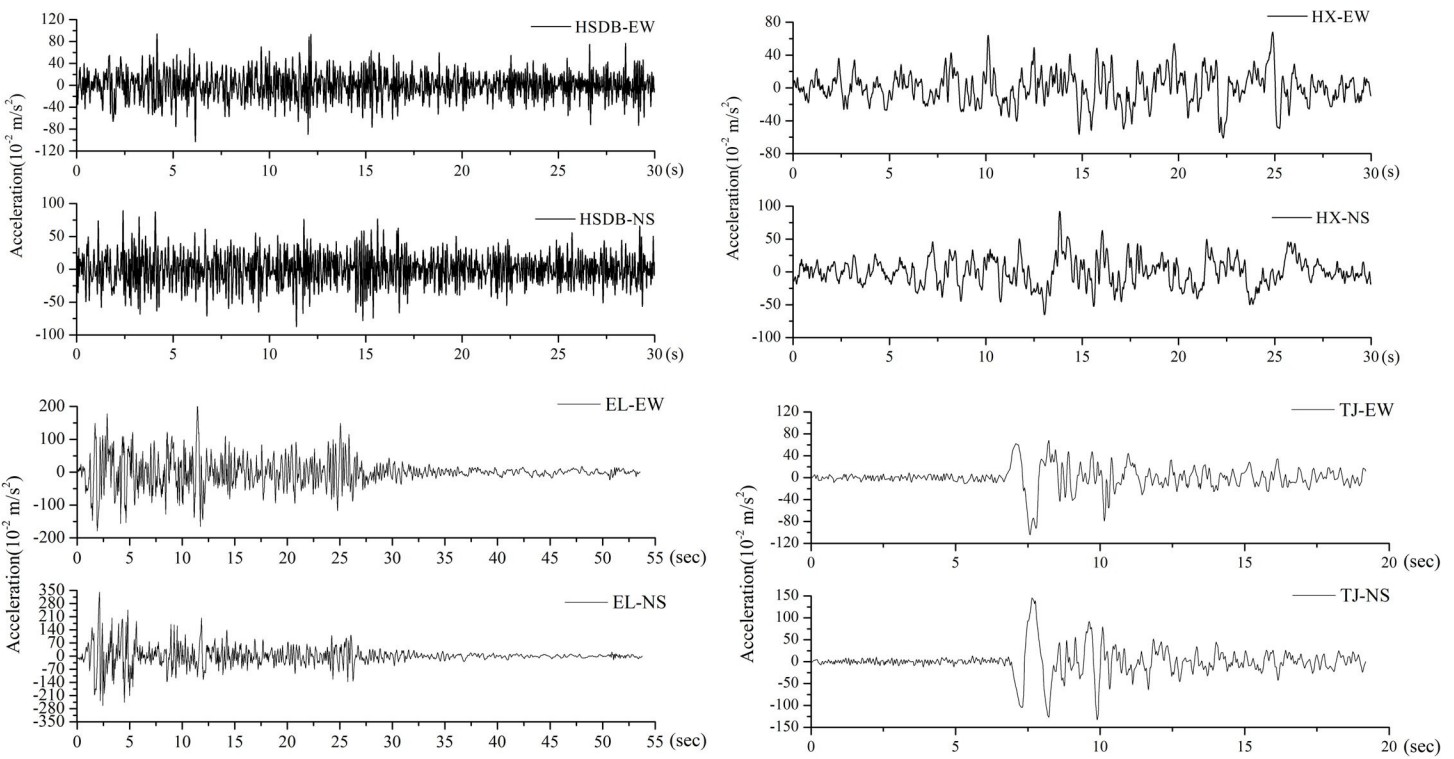

**Fig 7. Time-history curves of two seismic ground motions.** a) HSDB seismic ground motion. b) HX seismic ground motion. c) El Centro seismic ground motion. d) Tianjin seismic ground motion.

The response spectra of nine main ground motions, the mean response spectrum of all ground motions, and the design response spectrum are plotted in Fig 8. Among them, the design response spectrum is the long-period seismic design spectrum when $T_g$ is equal to 0.55 seconds in Fig 1. The mean response spectrum in Fig 8 is the 95% guarantee rate. The mean response spectrum obtained from nine ground motions is larger than the design response spectrum, mainly because the number of ground motions is too small. The design response spectrum mainly considers the safety of structure design and takes into account the economic and cost factors.

## 4 Results

### 4.1 Dynamic characteristic

The tank models analyzed in this paper are the models in Section 3.4. The first-order sloshing period of the tank liquid-solid coupling system is mainly used in the hydrodynamic pressure analyses. The brief tank model features and their dynamic characteristics are shown in Table 7. The first-order sloshing vibration modes of the six working conditions including A-10%, A-30%, A-50%, A-70%, B-70%, C-70% are shown in Fig 9. The following conclusions can be drawn from Table 7 and Fig 9. When the tank radius is the same, the sloshing period decreases with the increase of water storage height. When the water storage height is the same, the sloshing period increases with the increase of the tank radius. The storage height and tank radius have little effect on the vibration mode of liquid sloshing, and the vibration mode of liquid sloshing of the six working conditions in Fig 9 is basically the same.

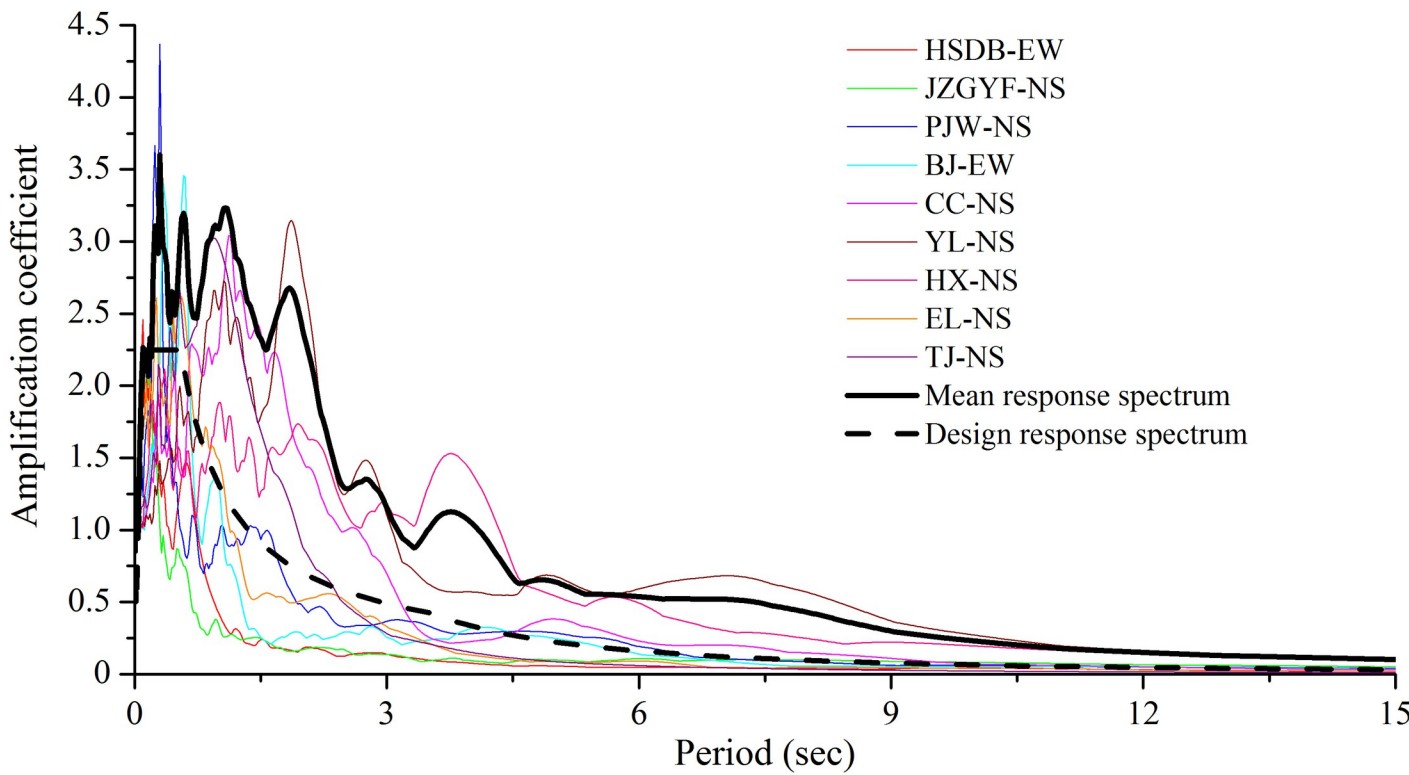

**Fig 8. Response spectrums, mean response spectrum, and design response spectrum of earthquakes.**

## 4.2 Hydrostatic pressure

In order to verify the correctness of the numerical simulation calculation results, the hydrostatic pressure is calculated by taking the A-50% condition as an example. The calculation formula of hydrostatic pressure is Eq (18).

$$p_{static} = \rho g z \tag{18}$$

Where, $p_{static}$ is hydrostatic pressure. The parameter $\rho$ is the density of the liquid. The density of water is 1000kg/m$^3$. The parameter $g$ is the acceleration of gravity, and its value is 9.81m/s$^2$. The parameter $z$ is the depth of the extraction point.

The locations of extraction points on the tank wall and bottom for the hydrodynamic and hydrostatic pressure are shown in Fig 10. For the working condition of A-50%, the $z$ value at the liquid surface position is 0, so the hydrostatic pressure is 0. The $z$ value of position 1 on the

**Table 7. Brief tank model features and dynamic characteristics results.**

| Working condition | Tank materials | Tank inner radius (m) | Water storage heights (m) | First-order sloshing period (s) | Mass participation percentage of the first-order sloshing period (%) |
|---|---|---|---|---|---|
| **A-10%** | Reinforced concrete | 6.75 | 0.35 | 12.201 | 15.75 |
| **A-30%** | | 6.75 | 1.05 | 7.128 | 33.34 |
| **A-50%** | | 6.75 | 1.75 | 5.650 | 41.55 |
| **A-70%** | | 6.75 | 2.45 | 4.929 | 44.84 |
| **B-70%** | | 4.3 | 2.45 | 3.368 | 34.07 |
| **C-70%** | | 13.5 | 2.45 | 9.461 | 55.08 |

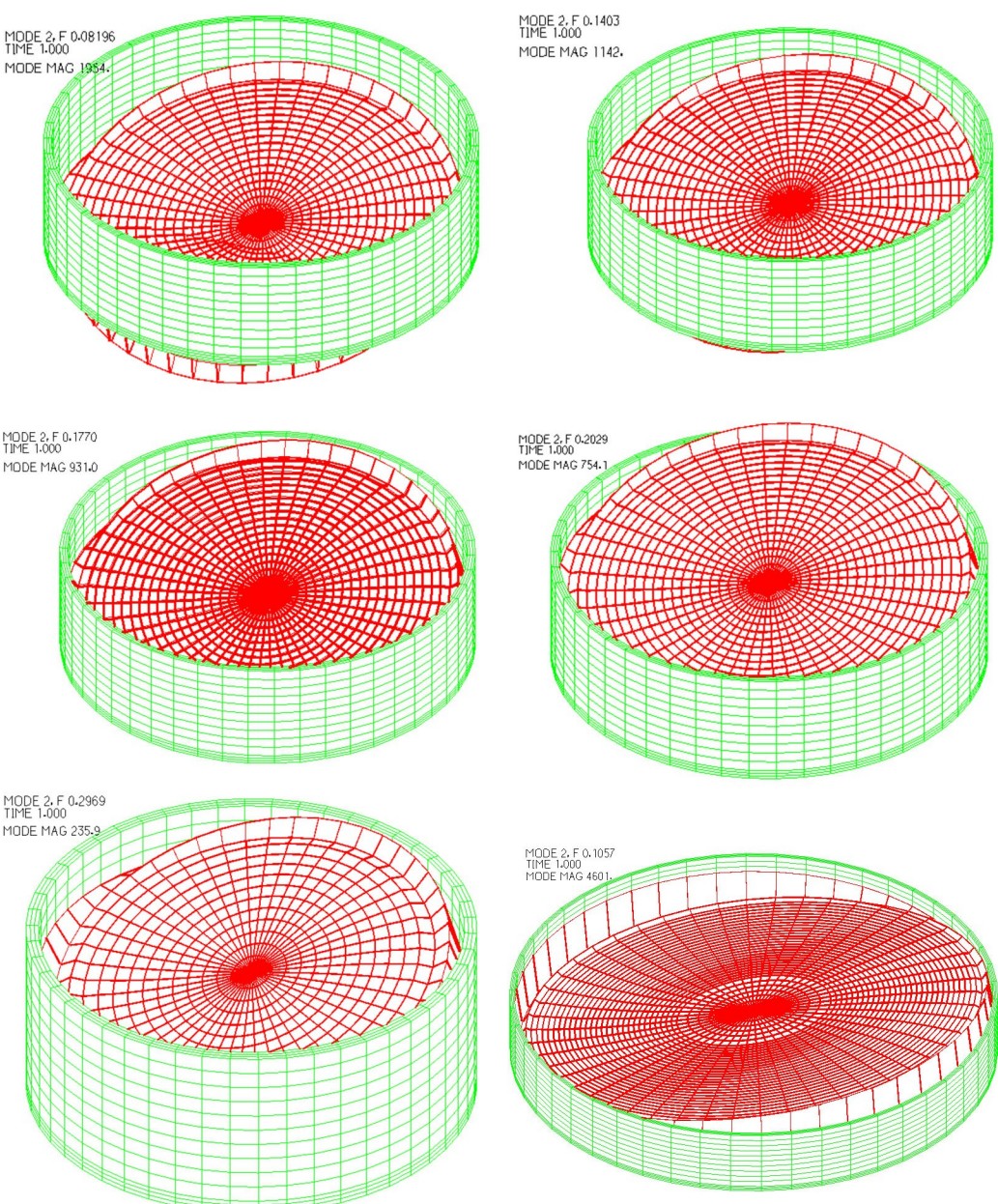

**Fig 9. Time-history curves of two seismic ground motions.** a) A-10%. b) A-30%. c) A-50%. d) A-70%. e) B-70%. f) C-70%.

tank wall is 0.35 meters, and the hydrostatic pressure was equal to the multiplication of parameters $\rho$, $g$ and $z$, that is, 1000 kg/m$^3$ multiplied by 9.81 m/s$^2$ multiplied by 0.35 meters equal to 3433.5 Pa. The hydrostatic pressure at other extraction points is calculated by this analogy. The calculation results are listed in Column 2 of Table 8. ADINA calculation model can be seen in Section 3.4. Static mode is selected when calculating hydrostatic pressure, and Dynamic-Implicit mode is selected when calculating hydrodynamic pressure. After static calculation in ADINA, the pressure at the corresponding position is extracted as the hydrostatic pressure, and the results are listed in Column 3 of Table 8. For Dynamic-Implicit calculation in ADINA, the pressure at the corresponding position is extracted as the total pressure, the hydrodynamic

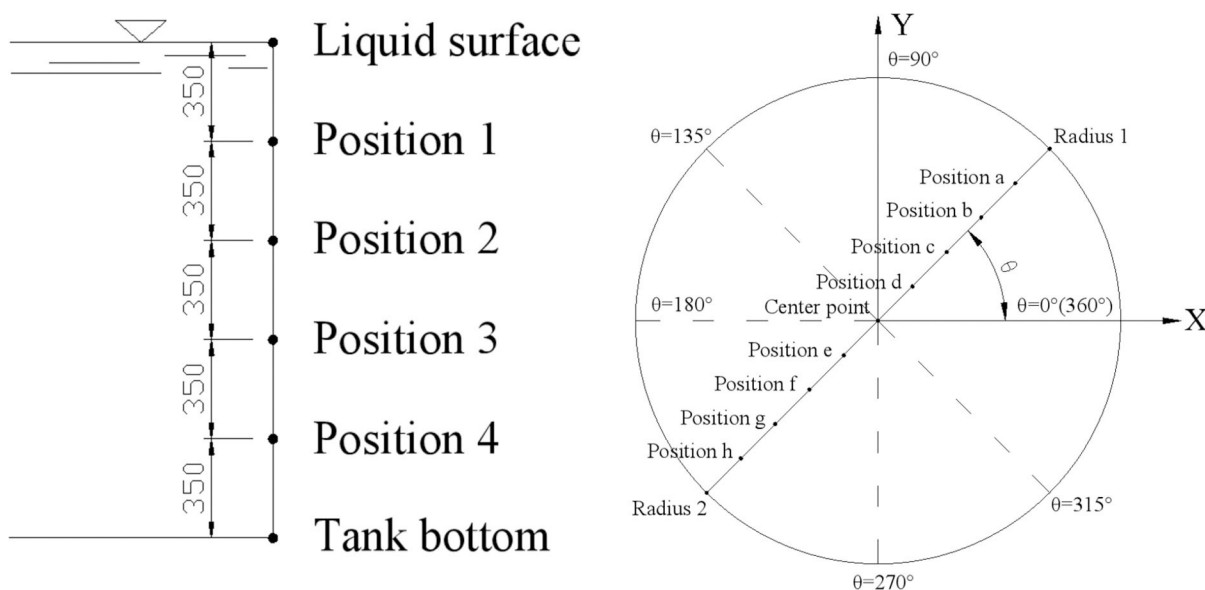

**Fig 10. Diagram showing position of extraction points for A-50% condition.** a)Points on the tank wall. b) Points in the circumferential and radial directions of the tank bottom.

pressure at a point is equal to the total pressure minus the hydrostatic pressure at this point. Therefore, it is necessary to calculate the hydrostatic pressure before calculating the hydrodynamic pressure at each point.

From the calculation results in Table 8, it can be seen that the hydrostatic pressures calculated theoretically and by ADINA are the same, so the hydrodynamic calculations of ADINA software are accurate and meet engineering requirements.

## 4.3 Hydrodynamic pressure under unidirectional horizontal seismic action

**4.3.1 Calculating results.** By comparing the hydrodynamic pressure ratios of each working condition under different peak accelerations for the same ground motion, it can be seen that the hydrodynamic pressure of the remaining points in addition to the circumferential points at 90 degrees and 270 degrees and the center point of the tank bottom at 2.0 m/s² peak acceleration is 1.9 to 2 times that at 1.0 m/s² peak acceleration, and that at 4.0 m/s² peak acceleration is 3.4 to 4 times that at 1.0 m/s² peak acceleration. The above analyses results show that the hydrodynamic pressure is related to the peak acceleration of the input ground motion, which is approximately proportional to the seismic coefficient $k$. At the same time, the hydrodynamic pressure under 1.0 m/s² peak acceleration multiplied by the seismic coefficient $k$ can be used to estimate the hydrodynamic pressure at other peak accelerations, which lie on the

**Table 8. Comparison of calculation results for hydrostatic pressure.**

| Extraction point | The hydrostatic values of theory calculations (Pa) | The hydrostatic values of ADINA calculations (Pa) |
|---|---|---|
| Liquid surface | 0 | -0.1378 |
| Position 1 | 3433.5 | 3433.4 |
| Position 2 | 6867.0 | 6866.9 |
| Position 3 | 10300.5 | 10300.4 |
| Position 4 | 13734.0 | 13733.9 |
| All points in the circumferential and radial directions | 17167.5 | 17167.4 |

safe side. Therefore, only the maximum values of the hydrodynamic pressures at the tank wall or the radial and circumferential directions of the tank bottom are considered. Taking A-50% working condition as an example, Tables 9–11 respectively list the maximum hydrodynamic pressure of tank wall, tank bottom radial direction, and circumferential direction under unidirectional main seismic motion with 1.0 m/s² peak acceleration. Due to a large amount of data, the data results of other working conditions are listed in the S1 to S15 Tables of the S1 File. The maximum distribution maps of the hydrodynamic pressures for all working conditions at 1.0 m/s² peak acceleration are given in this section in Figs 11–13, respectively.

It can be seen from Figs 11–13 that the hydrodynamic pressure distributions at the tank wall and in the radial and circumferential directions on the tank bottom are basically the same despite differences in the water storage heights. The hydrodynamic pressure is the maximum at the circumferential point of the tank bottom along the seismic input direction, and the hydrodynamic pressure is close to zero at the circumferential point of the tank bottom vertical to the seismic input direction. The distribution of the hydrodynamic pressure along the height direction of the tank wall is close to rectangular. The hydrodynamic pressure at the free liquid surface is greatest under long-period seismic motion action, and the hydrodynamic pressure at the tank bottom is greatest under short-period seismic motion action. The hydrodynamic pressure at the inner point of the tank bottom is lower than that at the radius point in the circumferential direction. From the radius point to the center point of the tank bottom, the hydrodynamic pressure exhibits an approximately linear distribution when the water storage height is higher. The hydrodynamic pressure at the one-half radius is high when the water storage height is lower and long-period ground motion inputs. The hydrodynamic pressure over the entire tank bottom is the maximum at the circumferential radius point. The hydrodynamic pressure at the inner point of the tank bottom is designed to be safe with respect to the hydrodynamic pressure at the circumferential radius point.

**4.3.2 Results comparison of the numerical, theoretical, and code methods.** Taking A-50% condition as an example, the hydrodynamic pressures of numerical simulation calculation, theoretical calculation and code calculation are compared. A comparison of the theoretical and numerical calculation results in Figs 11C) and 12c) show that the hydrodynamic pressures at the tank wall and tank bottom are more similar under short-period seismic motion action. The numerical calculation results of the hydrodynamic pressure at the tank wall exhibit the characteristics of a low liquid level and large tank bottom, and those at the tank bottom show distribution characteristics of an inverse tangent function near the origin. With an increase in the long-period components in the input ground motion, the numerical results become significantly higher than the theoretical results, and the maximum ratio of the two reaches 2.6. The numerical calculation results for the hydrodynamic pressure at the tank wall show the distribution characteristics of a high liquid level and small tank bottom, and those at the tank bottom show the distribution characteristics of a *sin* function near the origin.

**Table 9. The maximum hydrodynamic pressure of the tank wall under unidirectional main seismic motion with 1.0 m/s² peak acceleration for A-50% condition (KPa).**

| Position | HSDB-EW | JZGYF-NS | PJW-NS | BJ-EW | CC-NS | YL-NS | HX-NS | EL-NS | TJ-NS |
|---|---|---|---|---|---|---|---|---|---|
| Liquid surface | 0.822 | 1.626 | 3.656 | 2.702 | 4.282 | 5.615 | 7.124 | 1.918 | 1.239 |
| Position 1 | 1.598 | 1.858 | 3.505 | 2.753 | 3.943 | 5.163 | 6.733 | 1.749 | 1.399 |
| Position 2 | 2.351 | 2.171 | 3.624 | 2.804 | 3.675 | 4.755 | 6.506 | 2.355 | 1.878 |
| Position 3 | 2.808 | 2.429 | 3.698 | 2.868 | 3.642 | 5.026 | 6.392 | 2.901 | 2.380 |
| Position 4 | 3.067 | 2.573 | 3.739 | 3.079 | 3.656 | 5.300 | 6.333 | 3.220 | 2.692 |
| Tank bottom | 3.151 | 2.618 | 3.752 | 3.151 | 3.663 | 5.394 | 6.315 | 3.329 | 2.805 |

**Table 10. The maximum hydrodynamic pressure of the radial direction for tank bottom under unidirectional main seismic motion with 1.0 m/s² peak acceleration for A-50% condition (KPa).**

| Position | HSDB-EW | JZGYF-NS | PJW-NS | BJ-EW | CC-NS | YL-NS | HX-NS | EL-NS | TJ-NS |
|---|---|---|---|---|---|---|---|---|---|
| Radius 1 | 2.590 | 2.618 | 3.534 | 3.083 | 3.663 | 4.316 | 6.315 | 3.329 | 2.552 |
| Position a | 0.973 | 1.384 | 2.511 | 2.035 | 3.188 | 4.117 | 5.531 | 1.138 | 0.543 |
| Position b | 0.524 | 1.030 | 2.080 | 1.745 | 3.234 | 3.720 | 5.546 | 0.716 | 0.358 |
| Position c | 0.359 | 0.762 | 1.600 | 1.432 | 2.660 | 1.907 | 4.390 | 0.664 | 0.641 |
| Position d | 0.170 | 0.351 | 0.823 | 0.713 | 1.325 | 1.319 | 1.905 | 0.471 | 0.530 |
| Center point | 0.018 | 0.018 | 0.018 | 0.018 | 0.018 | 0.018 | 0.018 | 0.018 | 0.066 |
| Position e | 0.137 | 0.437 | 0.705 | 0.614 | 1.210 | 1.217 | 1.604 | 0.477 | 0.525 |
| Position f | 0.322 | 0.852 | 1.630 | 1.289 | 1.826 | 2.169 | 3.190 | 0.778 | 0.636 |
| Position g | 0.518 | 1.047 | 2.283 | 1.682 | 2.074 | 2.642 | 3.574 | 0.783 | 0.456 |
| Position h | 1.030 | 1.401 | 2.811 | 2.098 | 2.844 | 3.014 | 4.769 | 1.028 | 0.891 |
| Radius 2 | 3.151 | 2.390 | 3.752 | 3.151 | 3.519 | 5.394 | 5.891 | 3.307 | 2.805 |

The maximum hydrodynamic pressure of the X-axis forward direction for A-50% condition is 0.687 KPa by using the Chinese code GB50032-2003 [31] method in Section 3.3. Comparing the Chinese code GB50032-2003 and the numerical calculation results in Figs 11C) and 13, it can be seen that the code values are obviously small, and the numerical results are almost symmetrical along the vertical to the seismic input direction. The hydrodynamic pressure at the tank bottom is least along the vertical to the seismic input direction. From here, the hydrodynamic pressure at each point gradually increases along an arc and reaches a maximum along with the directions of the ground motion input. The above analyses show that the existing theoretical method for calculating hydrodynamic pressure cannot meet user demand and is unsafe under most working conditions. Therefore, a more accurate method for calculating hydrodynamic pressure must be established.

**4.3.3 Comparing the influence of water storage height.** To compare the influence of water storage height on hydrodynamic pressure under the same ground motion input, Fig 14

**Table 11. The maximum hydrodynamic pressure of the circumferential direction for tank bottom under unidirectional main seismic motion with 1.0 m/s² peak acceleration for A-50% condition (KPa).**

| Angle ($\theta°$) | HSDB-EW | JZGYF-NS | PJW-NS | BJ-EW | CC-NS | YL-NS | HX-NS | EL-NS | TJ-NS |
|---|---|---|---|---|---|---|---|---|---|
| 0°(360°) | 2.590 | 2.618 | 3.534 | 3.083 | 3.663 | 4.316 | 6.315 | 3.329 | 2.552 |
| 22.5° | 2.467 | 2.493 | 3.366 | 2.937 | 3.489 | 4.108 | 6.009 | 3.171 | 2.431 |
| 45° | 1.998 | 2.015 | 2.724 | 2.378 | 2.825 | 3.314 | 4.845 | 2.568 | 1.969 |
| 67.5° | 0.952 | 0.954 | 1.297 | 1.134 | 1.346 | 1.570 | 2.268 | 1.224 | 0.940 |
| 90° | 0.163 | 0.163 | 0.162 | 0.162 | 0.162 | 0.161 | 0.163 | 0.161 | 0.515 |
| 112.5° | 1.165 | 0.884 | 1.381 | 1.167 | 1.288 | 1.998 | 2.001 | 1.224 | 1.037 |
| 135° | 2.435 | 1.847 | 2.896 | 2.436 | 2.713 | 4.171 | 4.469 | 2.556 | 2.168 |
| 157.5° | 3.003 | 2.279 | 3.576 | 3.004 | 3.353 | 5.143 | 5.596 | 3.153 | 2.674 |
| 180° | 3.151 | 2.390 | 3.752 | 3.151 | 3.520 | 5.394 | 5.891 | 3.307 | 2.805 |
| 202.5° | 3.001 | 2.277 | 3.573 | 3.002 | 3.351 | 5.139 | 5.592 | 3.150 | 2.672 |
| 225° | 2.430 | 1.843 | 2.890 | 2.431 | 2.708 | 4.163 | 4.460 | 2.551 | 2.163 |
| 247.5° | 1.518 | 1.151 | 1.801 | 1.519 | 1.683 | 2.601 | 2.679 | 1.594 | 1.351 |
| 270° | 0.392 | 0.297 | 0.459 | 0.393 | 0.423 | 0.670 | 0.587 | 0.412 | 0.522 |
| 292.5° | 0.648 | 0.646 | 0.882 | 0.772 | 0.916 | 1.065 | 1.521 | 0.833 | 0.640 |
| 315° | 1.779 | 1.793 | 2.425 | 2.118 | 2.515 | 2.945 | 4.303 | 2.286 | 1.754 |
| 337.5° | 2.350 | 2.373 | 3.205 | 2.797 | 3.322 | 3.908 | 5.717 | 3.020 | 2.315 |

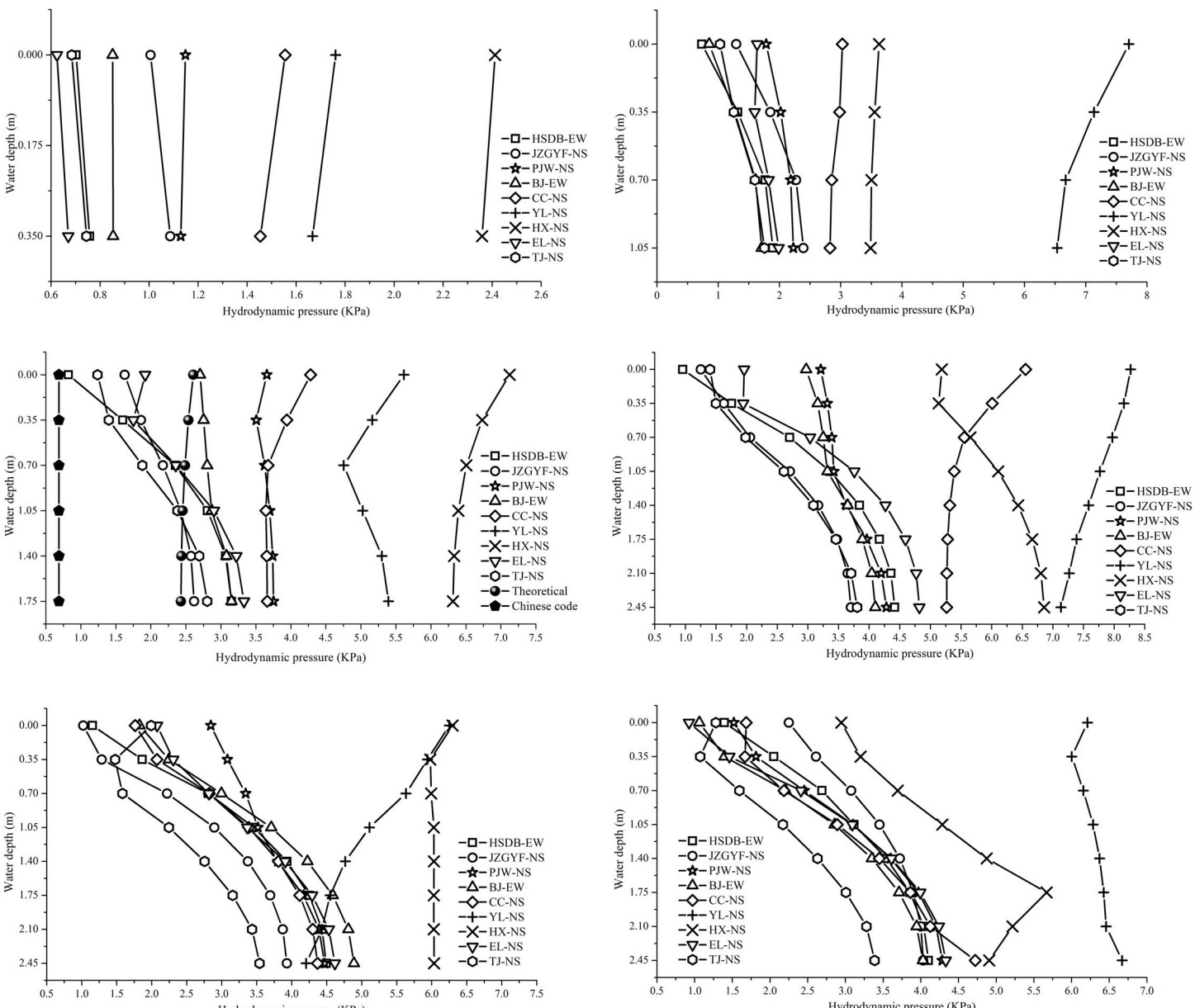

**Fig 11. Maximum hydrodynamic pressure distributions at the tank wall at 1.0 m/s$^2$ peak acceleration.** a) A-10% condition. b) A-30% condition. c) A-50% condition. d) A-70% condition. e) B-70% condition. f) C-70% condition.

shows a plot of the maximum hydrodynamic pressure distributions of the tank wall at different water storage heights under the same ground motion input with a 1.0 m/s$^2$ peak acceleration.

It can be seen from Fig 14 that under short-period seismic motion actions, the maximum hydrodynamic pressures at the liquid level are similar and low at different water storage heights. With an increase in the water storage height, the hydrodynamic pressure increases rapidly, and the hydrodynamic pressure at the tank bottom is much greater than that at the liquid level. Under medium and long-period ground motion actions, the maximum hydrodynamic pressures of the liquid level differ at different water storage heights. With an increase in the water storage height, the hydrodynamic pressure increases slowly, and the hydrodynamic

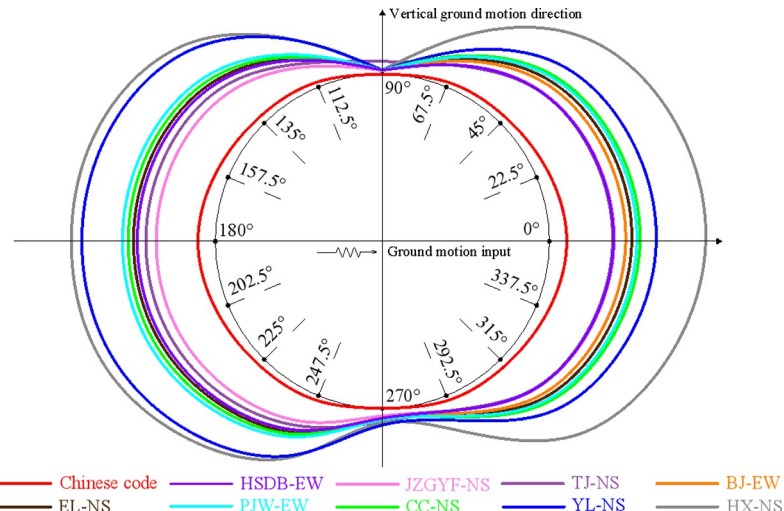

**Fig 13. Maximum hydrodynamic pressure distributions in the circumferential direction at the tank bottom at 1.0 m/s² peak acceleration for the A-50% condition.**

pressure at the tank bottom is not much different from that at the liquid level. Under long-period seismic motion actions, due to the substantial increase in the convective pressure, the hydrodynamic pressure at the liquid level is greater than that at the tank bottom in most cases. According to Fig 14, the following conclusions can be drawn: the hydrodynamic pressure distribution shape and values at the tank wall are different when the water storage heights differ. The water storage height is a major factor affecting the hydrodynamic pressure. Whatever the working condition, the hydrodynamic pressure differs little in the range of 0.5 m from the bottom of the tank.

**4.3.4 Comparing the influence of tank radius.** To compare the influence of the tank radius on the hydrodynamic pressure, Fig 15 shows a plot of the maximum hydrodynamic pressure distributions at the tank wall for tanks A, B, and C with a water storage height of 2.45 m under the same ground motion action with a 1.0 m/s² peak acceleration.

It can be seen from Fig 15 that the distribution of hydrodynamic pressure at the tank wall is similar for the same water storage height and different tank radii under short-period seismic motion actions and that the hydrodynamic distribution at the tank wall follows no obvious law under long-period seismic motion actions. The water storage height and the tank radius together determine the basic frequency of the liquid sloshing, which in turn affects the distributions of the convective and impulsive pressures.

**4.3.5 Calculation method.** From the above comprehensive analyses, it can be concluded that the hydrodynamic pressure at the radius point in the circumferential direction is greater than that at the inner point of the tank bottom and that the hydrodynamic pressure at the corresponding tank wall is also greatest when the radius point at $\theta$ angle of the tank bottom has maximum hydrodynamic pressure. Therefore, the hydrodynamic pressure distribution along the height of the tank wall at the radius point with the maximum hydrodynamic pressure can be used to characterize the maximum hydrodynamic pressure of the tank under certain ground motion actions. On this basis, in the following analyses, the hydrodynamic pressure distribution of the tank wall at the radius point with the maximum hydrodynamic pressure is taken as the focus of analysis and main research object to ensure that a reasonable calculation formula for hydrodynamic pressure is provided.

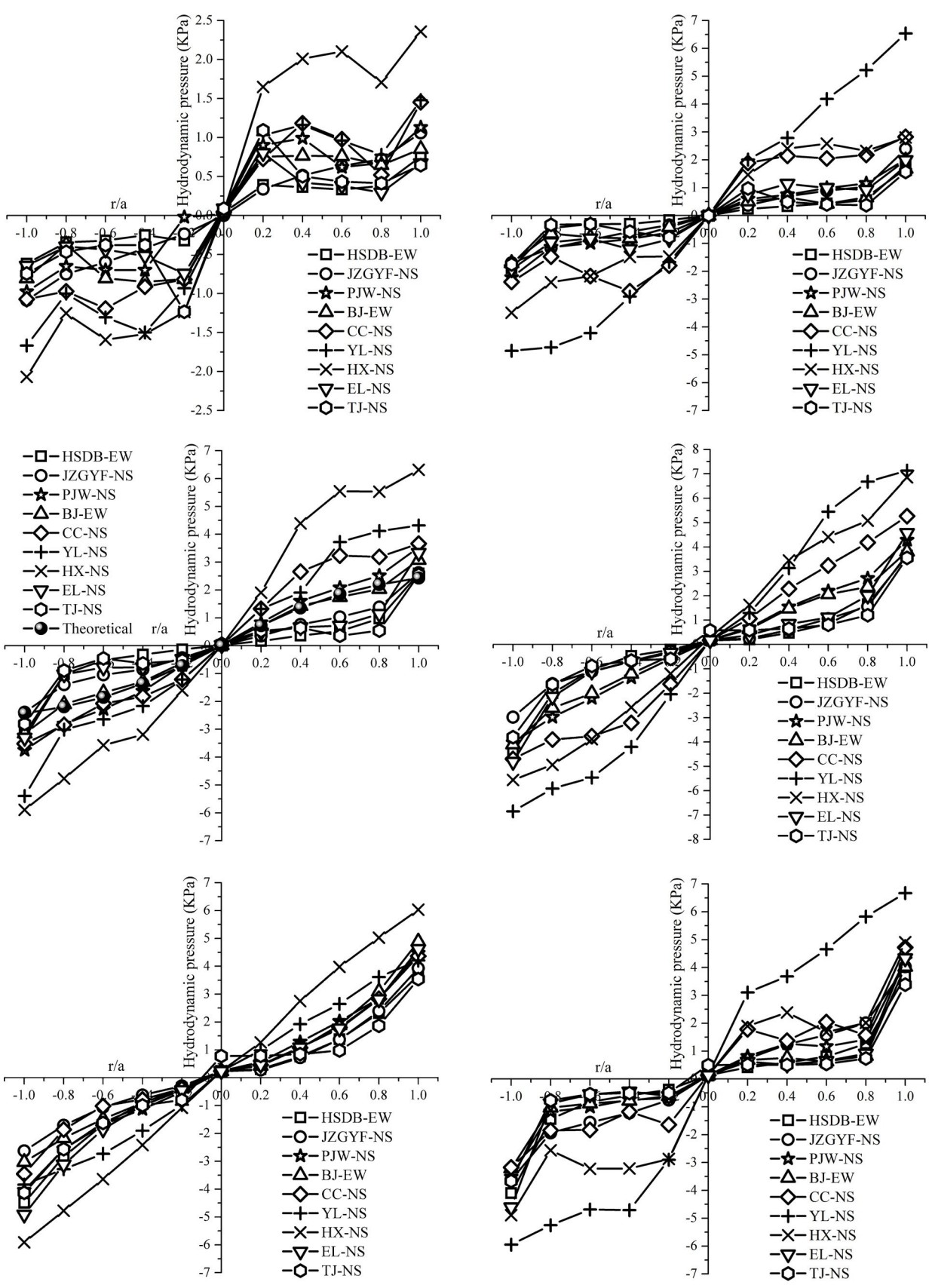

**Fig 12. Maximum hydrodynamic pressure distributions of the tank bottom at 1.0 m/s² peak acceleration.** a) A-10% condition. b) A-30% condition. c) A-50% condition. d) A-70% condition. e) B-70% condition. f) C-70% condition.

In summary, the hydrodynamic pressure is obtained by combining the square root of the sum of squares of the convective and impulsive pressures, as shown in Eq (19):

$$P_{WD} = \sqrt{P_{WS}^2 + P_{WP}^2} \qquad (19)$$

where $P_{WD}$ is the hydrodynamic pressure, $P_{WS}$ is the convective pressure, and $P_{WP}$ is the impulsive pressure.

Based on the shape and distribution characteristics obtained by the theoretical calculation formula for the convective pressure, the convective pressure can be expressed as shown in Eq (20), and the unit is KPa:

$$P_{WS} = c\gamma_w h \qquad (20)$$

where $c$ is the structural influence coefficient, 0.5 is taken for the reinforced concrete tank

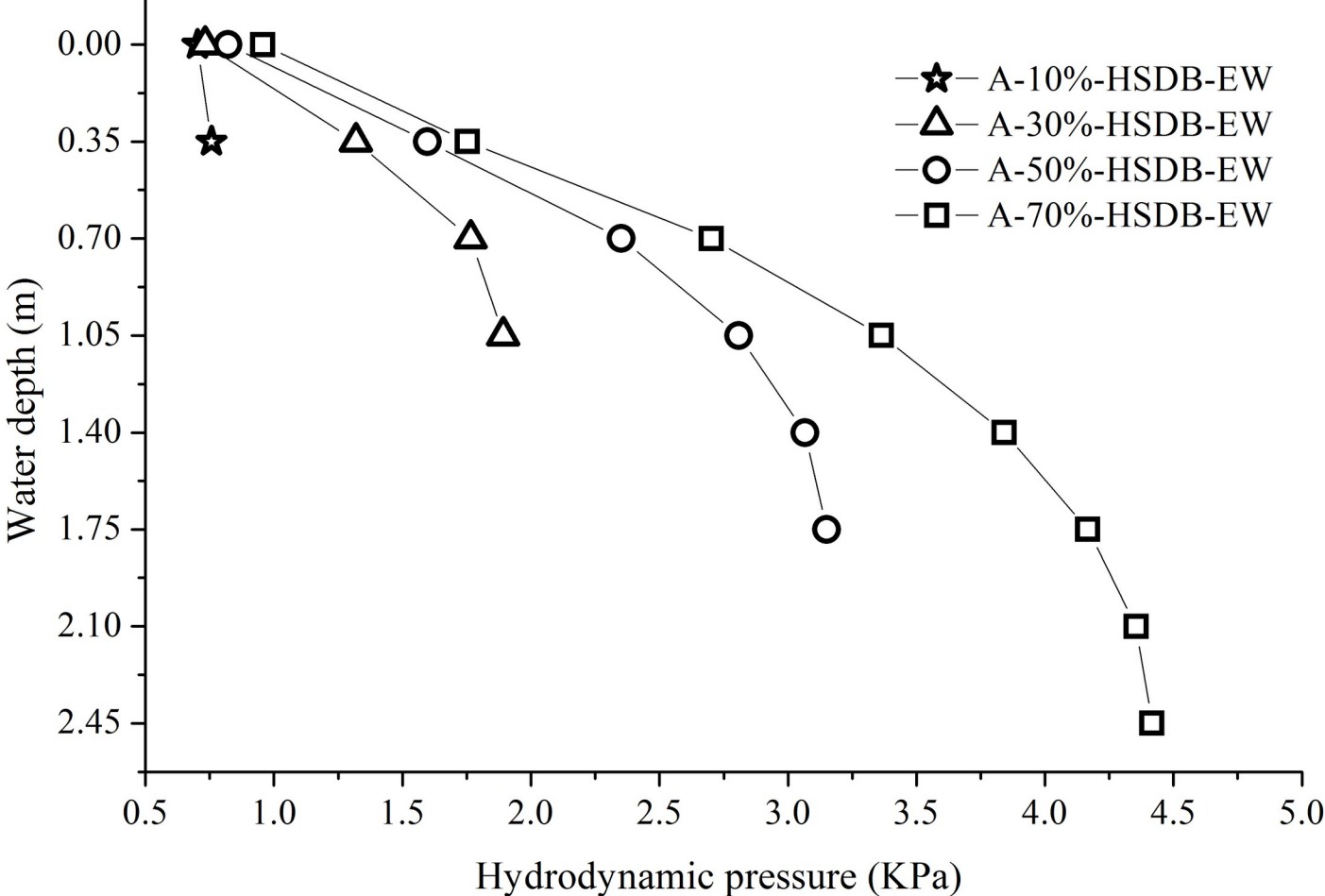

**Fig 14. Maximum hydrodynamic pressure distributions of a tank wall at different water storage heights under the same ground motion input with 1.0 m/s² peak acceleration.** a) HSDB-EW seismic motion. b) JZGYF-NS seismic motion. c) PJW-NS seismic motion. d) BJ-EW seismic motion. e) CC-NS seismic motion. f) YL-NS seismic motion. g) HX-NS seismic motion. h) EL-NS seismic motion. i) TJ-NS seismic motion.

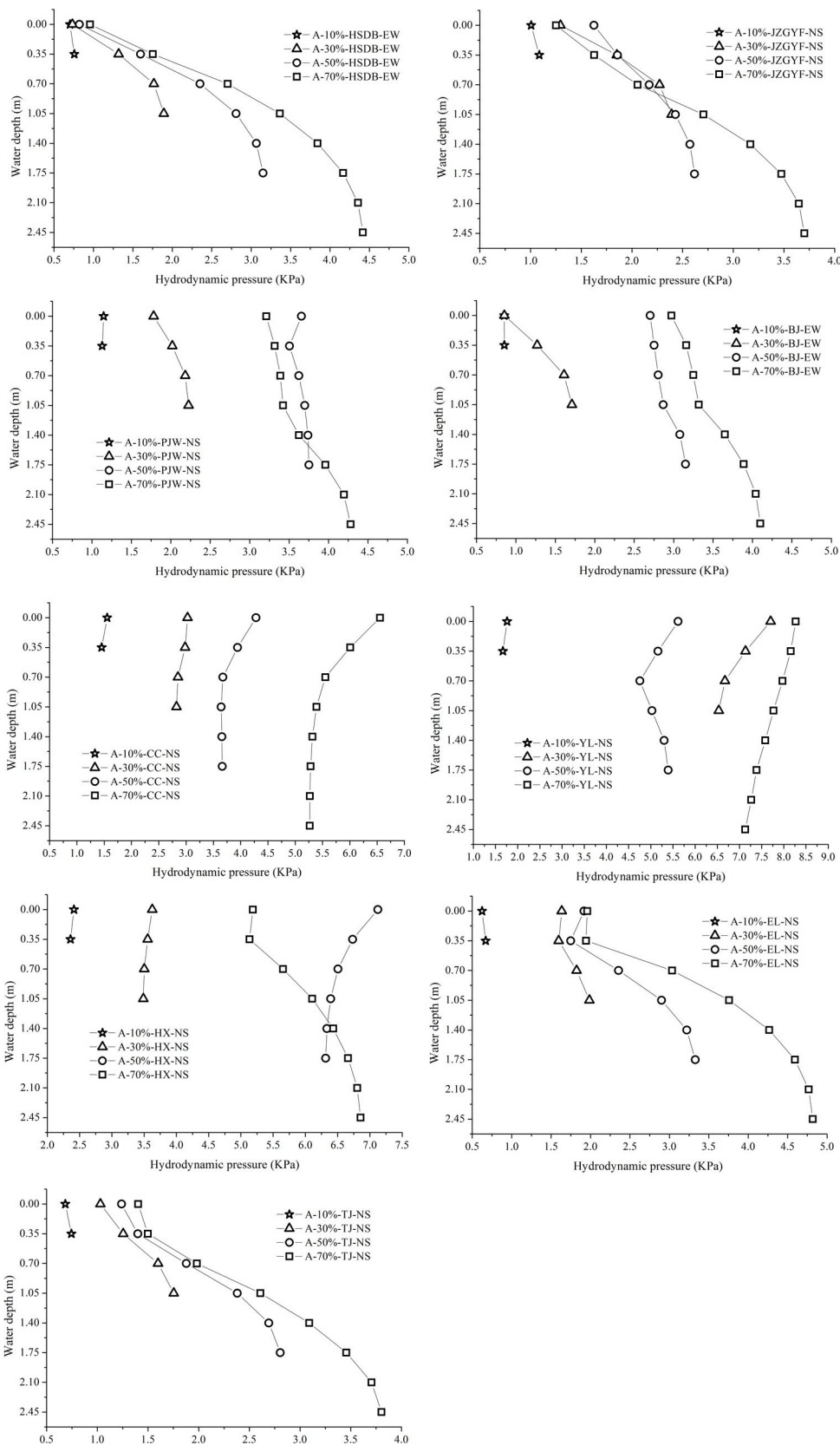

**Fig 15. Maximum hydrodynamic pressure distributions of the tank wall at different tank radii input with 1.0 m/s²
peak acceleration.** a) HSDB-EW seismic motion. b) JZGYF-NS seismic motion. c) PJW-NS seismic motion. d) BJ-EW
seismic motion. e) CC-NS seismic motion. f) YL-NS seismic motion. g) HX-NS seismic motion. h) EL-NS seismic
motion. i) TJ-NS seismic motion.

structure, and 0.7 is taken for the unreinforced brick masonry tank structure. $\gamma_W$ is the bulk
density of water (kN/m³). $h$ is the maximum sloshing wave height of a liquid surface (m),
which is calculated as described in section 5.4.

The impulsive pressure is equally distributed along with the height of the tank wall, and the
unit is KPa. The following two calculation methods are established through the analyses.

The first calculation method is as shown in Eq (21):

$$P_{WP1} = ck\gamma_W R \cdot f_{w1} \tag{21}$$

where, the definitions of $c$ and $\gamma_W$ are the same as in Eq (20). The parameter $k$ is a seismic coef-
ficient, $R$ is the inner radius of the tank (m), and $f_{w1}$ is the coefficient of impulsive hydrody-
namic pressure. The values for the coefficient of impulsive hydrodynamic pressure $f_{w1}$
corresponding to the values for $H_W/R$ are shown in Table 12.

The second calculation method is as shown in Eq (22):

$$P_{WP2} = 2.1 \cdot k \cdot \gamma_W \cdot H_W \tag{22}$$

where, $H_W$ is the water storage height (m). The definitions of $k$ and $\gamma_W$ can be seen the Eqs (1)
and (20), respectively.

The hydrodynamic pressure at the circumferential point of the bottom plate under unidi-
rectional horizontal seismic action is related to its position. The positive direction along the
seismic action is 0 degrees, and the counterclockwise angle is continuously increased. The dis-
tribution law of hydrodynamic pressure at the circumferential point $P_{WD}(\theta)$ is obtained using
the numerical calculation results. The hydrodynamic pressure at the tank bottom at the $\theta$
angle is calculated using Eq (23), and the unit is KPa.

$$\begin{cases} P_{0°} = P_{180°} = P_{WD} \\ P_{WD}(\theta) = P_{WD} \cdot |\cos\theta| + 0.85 \quad (\theta \neq 0°, \theta \neq 180°) \end{cases} \tag{23}$$

The hydrodynamic pressure distributions are finally obtained along with the height of the tank
wall and the circumferential direction of the tank bottom, as shown in Fig 16.

## 4.4 Hydrodynamic pressure under bi-directional horizontal seismic action

The hydrodynamic pressure still presents the distribution form with the maximum hydrody-
namic pressure at the circumferential direction of the tank bottom and the minimum hydro-
dynamic pressure at the central point of the tank bottom under the bi-directional horizontal
seismic action. For short-period seismic actions, the hydrodynamic pressure along the height
of the tank wall is the largest at the tank bottom. For long-period seismic actions, the hydrody-
namic pressure along the height of the tank wall is the largest at the liquid surface, and the bot-
tom is relatively small. Therefore, similar to the analysis of hydrodynamic pressure under

**Table 12. Coefficient of impulsive hydrodynamic pressure $f_{w1}$ for a ground-level circular tank.**

| $H_W/R$ | 0.1 | 0.2 | 0.3 | 0.4 | 0.5 | 0.6 |
|---------|-----|-----|-----|-----|-----|-----|
| $f_{w1}$ | 0.43 | 0.76 | 1.09 | 1.42 | 1.74 | 2.07 |

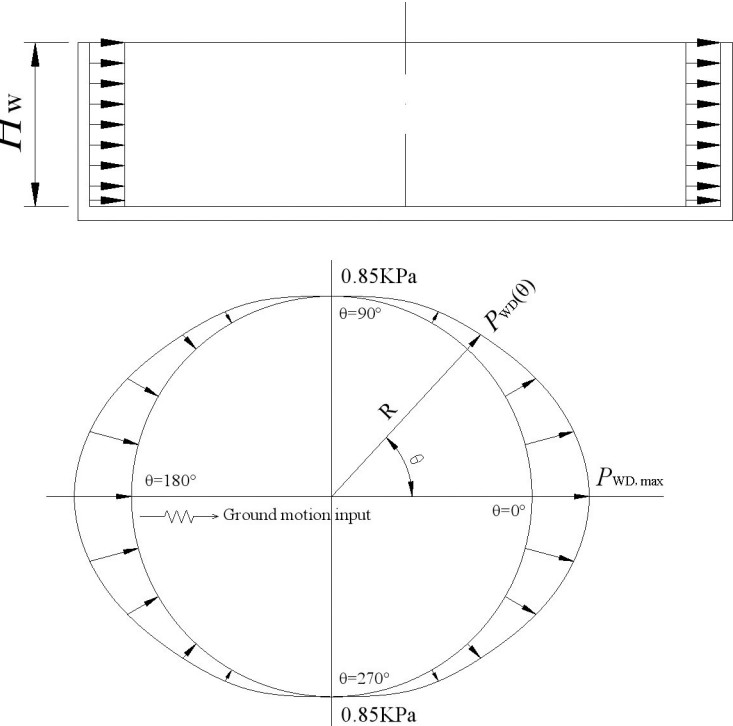

**Fig 16. Hydrodynamic pressure distributions of the ground rested circular RC tank.** a) Along the height of the tank wall. b) Along the circumferential direction of the tank bottom.

unidirectional horizontal seismic action, the hydrodynamic pressure under bi-directional horizontal seismic action can be represented by the hydrodynamic pressure distribution of the tank wall at the position of the maximum hydrodynamic pressure in the circumferential direction of the tank bottom. Tables 13 and 14 respectively list the maximum hydrodynamic pressure of the whole tank under unidirectional main and secondary seismic motion with 1.0 m/s² peak acceleration. Table 15 lists the maximum hydrodynamic pressure of the whole tank under bi-directional seismic motions with 1.0 m/s² peak acceleration.

By analyzing and comparing the calculation results of hydrodynamic pressure under unidirectional main seismic action input, unidirectional secondary seismic action input, and bi-directional seismic action input, the calculation formula of hydrodynamic pressure under bi-directional seismic action is fitted by direct addition, SRSS method, or reduced SRSS method. Three fitting methods are compared from the perspective of numerical reliability. SRSS method can ensure that the hydrodynamic pressure is reliable and the calculation is relatively

**Table 13. The maximum hydrodynamic pressure $P_{WD,x}$ of the whole tank under unidirectional main seismic motion with 1.0 m/s² peak acceleration (KPa).**

| Working condition | HSDB-EW | JZGYF-NS | PJW-NS | BJ-EW | CC-NS | YL-NS | HX-NS | EL-NS | TJ-NS |
|---|---|---|---|---|---|---|---|---|---|
| A-10% | 0.757 | 1.086 | 1.148 | 0.853 | 1.555 | 1.761 | 2.412 | 0.670 | 0.744 |
| A-30% | 1.892 | 2.391 | 2.226 | 1.710 | 3.028 | 7.404 | 3.926 | 1.986 | 1.754 |
| A-50% | 3.151 | 2.618 | 3.752 | 3.151 | 4.282 | 5.615 | 7.124 | 3.329 | 2.805 |
| A-70% | 4.417 | 3.701 | 4.280 | 4.099 | 6.555 | 8.268 | 6.856 | 4.819 | 3.804 |
| B-70% | 4.493 | 3.930 | 4.468 | 4.888 | 4.368 | 6.251 | 6.298 | 4.914 | 4.127 |
| C-70% | 4.100 | 4.031 | 4.284 | 4.028 | 4.722 | 6.670 | 5.269 | 4.612 | 3.667 |

**Table 14. The maximum hydrodynamic pressure $P_{WD,y}$ of the whole tank under unidirectional secondary seismic motion with 1.0 m/s$^2$ peak acceleration (KPa).**

| Working condition | HSDB-NS | JZGYF-EW | PJW-EW | BJ-NS | CC-EW | YL-EW | HX-EW | EL-EW | TJ-EW |
|---|---|---|---|---|---|---|---|---|---|
| A-10% | 0.783 | 0.906 | 1.636 | 0.669 | 0.884 | 1.140 | 0.867 | 0.905 | 0.533 |
| A-30% | 1.800 | 1.888 | 2.553 | 1.579 | 1.829 | 3.269 | 3.284 | 2.109 | 1.103 |
| A-50% | 3.079 | 3.255 | 4.688 | 2.868 | 3.913 | 5.005 | 7.190 | 3.938 | 1.781 |
| A-70% | 3.974 | 4.385 | 6.602 | 3.875 | 6.175 | 5.654 | 6.095 | 4.296 | 2.480 |
| B-70% | 3.759 | 3.795 | 4.527 | 3.130 | 4.274 | 4.908 | 5.483 | 4.230 | 2.420 |
| C-70% | 3.800 | 4.047 | 5.559 | 3.224 | 4.052 | 5.244 | 3.741 | 3.736 | 2.339 |

simple. The final calculation formula is shown in Eq (24).

$$P_{WD,Ek} = \sqrt{P_{WD,x}^2 + P_{WD,y}^2} \tag{24}$$

Where, $P_{WD,Ek}$ is the standard value of hydrodynamic pressure under bi-directional horizontal seismic action, and its unit is KPa. $P_{WD,x}$ and $P_{WD,y}$ are the standard values of hydrodynamic pressure under X and Y directional horizontal seismic actions, which are calculated according to the method in Section 6.5. And their units are all KPa.

## 5 Discussions

From results reported in the reference [12] and the theoretical calculation formulas for hydrodynamic pressure presented in Section 3.2, the following can be stated: hydrodynamic pressure is divided into two parts: the convective pressure generated by the liquid surface sloshing (caused by earthquakes of long-period displacement type) and the impulsive pressure generated by the liquid-solid coupled vibration (caused by earthquakes of short-period acceleration type). Regarding the superposition of the convective pressure and impulsive pressure, there are currently three treatment methods:

(1) The impulsive and convective pressures are directly added to design a liquid-containing structure in the seismic design of liquid-containing concrete structures (ACI 350.3–01) and commentary (350.3R-01) provision 4.1 [13], the Welded Steel Tanks for Oil Storage (US API 650) provision E.6.1 [14] and Welded Steel Tanks for Water Storage (AWWA D100) [15].

(2) The impulsive and convective pressures cannot be superimposed and should be evaluated separately, because the impulsive pressure caused by the displacement-type earthquake and the convective pressure caused by the acceleration-type earthquake are addressed in the welded steel tanks for oil storage (JIS B 8501–1985) [35] and other Japanese seismic specifications for oil storage tanks.

**Table 15. The maximum hydrodynamic pressure $P_{WD,2}$ of the whole tank under bi-directional seismic motions with 1.0 m/s$^2$ peak acceleration (KPa).**

| Working condition | HSDB2 | JZGYF2 | PJW2 | BJ2 | CC2 | YL2 | HX2 | EL2 | TJ2 |
|---|---|---|---|---|---|---|---|---|---|
| A-10% | 0.838 | 1.472 | 1.945 | 0.896 | 1.673 | 2.348 | 2.554 | 1.041 | 0.802 |
| A-30% | 1.939 | 2.475 | 2.888 | 1.711 | 3.500 | 8.087 | 5.010 | 2.744 | 1.845 |
| A-50% | 3.170 | 3.220 | 5.513 | 3.250 | 4.450 | 7.490 | 8.340 | 4.329 | 3.002 |
| A-70% | 4.516 | 4.886 | 7.778 | 4.225 | 7.266 | 9.470 | 7.411 | 5.183 | 4.084 |
| B-70% | 4.768 | 4.483 | 5.497 | 4.921 | 4.744 | 7.750 | 8.015 | 5.345 | 4.179 |
| C-70% | 4.317 | 5.460 | 6.729 | 4.049 | 4.956 | 8.471 | 6.754 | 4.936 | 3.991 |

(3) It is only necessary to check the impulsive pressure in the Chinese Seismic design code for outdoor supply water and drainage, gas, and thermal engineering (TJ32-78) [36], because the convective pressure is low and the liquid surface hardly sloshes at all under the action of ground motions recorded at predominant periods of 0.1–1 seconds. At the same time, the tank wall is regarded as an elastic system, for which the influence of deformation is taken into account, and the convective pressure of a certain effect is considered, so the distribution of the hydrodynamic pressure along the height of the tank wall is set to the equivalent pressure from top to bottom. At present, code GB50032-2003 [31] uses this regulation.

From the earthquake damage of a liquid-containing structure, it can be determined that the severe sloshing of the storage liquid during an earthquake would also cause liquid spillage in addition to structural damage. The structural vibration period is generally less than 0.1 s, and the liquid-containing sloshing period is usually in the range of 3–15 s. Since the difference between the two is very large, the two vibrations experience no substantial coupling effect, and the maximum value cannot occur at the time of an earthquake occurrence. When the impulsive pressure caused by the liquid-solid coupled vibration is calculated, the liquid surface sloshing can be neglected. That is, the influence of a gravity wave on the liquid surface can be ignored. At the same time, based on the theoretical calculation formula for the maximum hydrodynamic pressure at a tank wall and tank bottom, as presented in section 6.1, the convective and impulsive pressures can be combined to obtain the total hydrodynamic pressure by the square root of the sum of squares to calculate the internal force on the structure.

It can be seen from the theoretical calculation formula for the convective and impulsive pressures that the impulsive pressure is the maximum at the bottom of the tank, and gradually decreases from the bottom of the tank to the liquid level along with the height of the tank wall. The convective pressure is greatest at the liquid level, and gradually increases from the bottom of the tank to the liquid level. Over the entire tank bottom plate, the convective pressure reaches the maximum at the circumferential radius, and gradually decreases from the circumferential radius to the center point. The impulsive pressure gradually increases from the center point of the bottom plate to the circumferential radius, but is close to 0 near the one-half radius, and reaches the maximum at the circumferential radius. For tank structures with ratios of height to diameter less than 1, the convective pressure plays an important role whether the hydrodynamic pressure is greatest at the tank wall or the bottom plate. Therefore, the convective pressure cannot be ignored. It can also be seen from the theoretical calculation results that the convective pressure changes little from the tank bottom to the liquid level, and the convective pressure can be simplified as an equivalent distribution along with the height of the tank wall, with its value reaching maximum along with the height of the tank wall. The impulsive pressure can be similarly simplified based on its theoretical calculations. Thus, the distributions of the convective and impulsive pressures at the height of the tank wall are identical. That is, the hydrodynamic pressures along the height of the tank wall are equally distributed, which is also consistent with the finite element results.

## 6 Conclusions

This paper analyzes the hydrodynamic pressure of the ground-rested circular RC tank by combining theoretical analysis, numerical simulation, and code calculation. The following conclusions and research results are obtained.

(1) Convective pressure and impulsive pressure are important components of hydrodynamic pressure.

(2) Convective pressure is proportional to the water bulk density and the maximum sloshing wave height under unidirectional horizontal seismic action. Convective pressure can be simplified as equivalent distributions along with the height of the tank wall. The influence of convective pressure must be considered when the hydrodynamic pressure is calculated, whether for short-period or long-period seismic motion actions.

(3) Impulsive pressure is proportional to the water bulk density, the horizontal seismic coefficient, tank inner radius, and the water storage height under unidirectional horizontal seismic action. Impulsive pressure can also be simplified as equivalent distributions along with the height of the tank wall.

(4) Hydrodynamic pressure under unidirectional horizontal seismic action is obtained by combining convective pressure and impulsive pressure through the square root of the sum of the squares, which value is equivalent along with the height of the tank wall.

(5) Hydrodynamic pressure under bi-directional horizontal seismic action is obtained by combining X and Y unidirectional hydrodynamic pressure through the square root of the sum of the squares.

## Supporting information

**S1 File. The maximum hydrodynamic pressure of the tank wall under unidirectional main seismic motion with 1.0 m/s$^2$ peak acceleration.**
(DOCX)

## Acknowledgments

The authors would like to thank the copyrighted ADINA software provided by the Experiment Education Center of Civil Engineering in Chongqing University of Arts and Sciences.

## Author Contributions

**Conceptualization:** Lin Gao.

**Formal analysis:** Lin Gao.

**Funding acquisition:** Lin Gao.

**Investigation:** Lin Gao, Zengshun Chen, Huihui Yang, Xiaohui Wan.

**Methodology:** Lin Gao, Zengshun Chen, Huihui Yang, Xiaohui Wan.

**Project administration:** Lin Gao.

**Resources:** Lin Gao, Zengshun Chen, Huihui Yang, Xiaohui Wan.

**Software:** Lin Gao.

**Supervision:** Lin Gao.

**Writing – original draft:** Lin Gao.

**Writing – review & editing:** Lin Gao, Zengshun Chen, Huihui Yang, Xiaohui Wan, Mingzhen Wang.

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
