## [Decision Letter · Decision Letter 0]

30 Sep 2021

PONE-D-21-24114Hydrodynamic pressure law of ground-rested circular RC tank under bi-directional horizontal seismic actionPLOS ONE

Dear Dr. Wang,

Thank you for submitting your manuscript to PLOS ONE. After careful consideration, we feel that it has merit but does not fully meet PLOS ONE’s publication criteria as it currently stands. Therefore, we invite you to submit a revised version of the manuscript that addresses the points raised during the review process.There are some grammatical errors in the manuscript. Please recheck the English of the manuscript.The background and importance of the research should clearly discuss the importance of the research. A little literature is given. Several references such as ACI 350.3-06, API standard 650 (11th Edition), AWWA standard D100-96, (GB 50032-2003)”,and several more are not cited in the text. Please use cite them properly.Detail of Finite Element Modeling is missing. Even in reference 13, it seems incomplete.On what basis earthquakes are selected? Please provide the response spectrum of each earthquake, and the mean response spectrum of the earthquakes.What is the mass participation percentage of the modes in Table 1?Gal is not part of an International System of Units (SI). Please change the unit of acceleration to SI.

We look forward to receiving your revised manuscript.

Kind regards,

Ahad Javanmardi, Ph.D

Academic Editor

PLOS ONE

Journal Requirements:

This research is funded by the Chongqing Natural Science Foundation (CSTC2019jcyj- msxmX0739).

This research is funded by the Chongqing Natural Science Foundation (CSTC2019jcyj- msxmX0739).

This research is funded by the Chongqing Natural Science Foundation (CSTC2019jcyj- msxmX0739).

Additional Editor Comments:

When revising your manuscript, please address each reviewers' comment specifically and in detail, rather than just mention that it is corrected in the manuscript. Please outline every change made in response to their comments and provide suitable rebuttals for any comments not addressed.  Also, do identify where in the text you have made the changes by red color, it is particularly helpful to note the page and line numbers from the original manuscript and revision so comparisons can be made.

Reviewers' comments:

Reviewer's Responses to Questions

**Comments to the Author**

1. Is the manuscript technically sound, and do the data support the conclusions?

Reviewer #1: Yes

Reviewer #2: Yes

Reviewer #3: Partly

Reviewer #4: Yes

2. Has the statistical analysis been performed appropriately and rigorously? 

Reviewer #1: N/A

Reviewer #2: Yes

Reviewer #3: N/A

Reviewer #4: Yes

3. Have the authors made all data underlying the findings in their manuscript fully available?

Reviewer #1: No

Reviewer #2: Yes

Reviewer #3: No

Reviewer #4: Yes

4. Is the manuscript presented in an intelligible fashion and written in standard English?

Reviewer #1: Yes

Reviewer #2: Yes

Reviewer #3: No

Reviewer #4: Yes

5. Review Comments to the Author

Reviewer #1: - Add and specify numerical model in ADINA

- Correct text in chapter 5, Table lists the hydrostatic values, not hydrodynamic values.

- Add hydrodynamic values from numerical model in ADINA

- Correct arrows in Fig. 12

- Why is the non-typical pressure C-70-HX-NS IN Fig. 11b?

- Add comparison of individual solutions (numerical, theoretical, and code)

Reviewer #2: The study is useful but it seems incomplete. The variation in impulsive and convective pressure under the selected earthquake can not provide a conclusion. The convective component not only depends on liquid depth but also on the dominant frequencies of the earthquake. Hence, adding more earthquake to the investigation is necessary.

Reviewer #3: This paper presents a study on circular RC tanks under bi-directional shaking. While the topic is important, it is unclear what the novelty is. Many details regarding analysis and modelling are missing. Specific comments are below.

1. Abstract: Novelty in method, structure considered, and results is unclear.

2. Introduction: The gap in the state-of-the-art and specific questions addressed are not clearly identified in the Introduction section.

3. Table 1: Should it be “Tank model storage” instead of “Tank model shortage”?

4. Section 2: Complete details on the geometric and material properties of the tank should be provided, so that the results can be simulated by others. Necessary drawings should also be provided.

5. Fig. 2: Response spectra of ground motions should be compared with target response spectrum.

6. Section 4: What is the definition of predominant period?

7. Table 3 presents the results obtained using theoretical expressions and ADINA software. However, no detail on the theoretical expressions and software modelling are provided so far.

8. Eq. 2, 3: All parameters should be defined.

9. Section 6.1: It is unclear if an existing solution is used, or is it for a new structure and support condition, or if a new approach for formulation is used.

Reviewer #4: The paper attempts to determine the link between sloshing wave height and hydrodynamic pressure, as well as to verify the hydrodynamic pressure components and their combination. The paper developed a method for calculating hydrodynamic pressure and presented its distribution rules. I appreciate the author's efforts, and I have some comments that I believe may help to improve the current form of the article.

1)The paper's structure is unclear. The reader will have difficulty understanding the article in the present format; for example, the introduction takes the form of a literature review.

1-1) Please ensure that you have an introduction section that meets the specifications listed below:

-Indicate the work's field, why it is vital, and what has already been done with proper citations.

-Indicate a gap, raise a research question, or criticise previous work in this area.

-Outline the goal and explain the current study, emphasising what is unique and why it is significant.

1-2) for the literature review part, you can modify the present introduction to perform as a literature review.

1-3) You need a methodology part. The goal is to provide enough information so that a competent researcher can replicate the study. Many of your readers will skip this part since they already know the broad methods you employed from the Introduction. However, careful drafting of this part is required since your results must be repeatable in order to be scientifically valid. Otherwise, your paper is not scientifically sound.

1-4) Your article also lacks a section on results. Please make in accordance.

2) The result indicates that the paper's findings are a continuation of previous research such as Mingzhen, W., & Lin, G. (2018) [Dynamic Time-History analyses of Ground Reinforced Concrete Tank in Water Supply System under Bi-directional Horizontal Seismic Actions]. Table 2 and several other data from this study were used without citation. Please Make sure that you maintain the novelty and refrain from repeating the same outcome.

3) Please cite the primary sources of the formulas you utilised in this study.

I look forward to reading your revised version. Good Luck!

6. PLOS authors have the option to publish the peer review history of their article (what does this mean?). If published, this will include your full peer review and any attached files.

Reviewer #1: No

Reviewer #2: **Yes: **Amiya Ranjan Pandit

Reviewer #3: No

Reviewer #4: No

---

## [Author Response · Author response to Decision Letter 0]

8 Feb 2022

Dear editor,

We have substantially revised our manuscript submitted to PLOS ONE, An International Journal (Manuscript number: PONE-D-21-24114) after reading the editor’s and four reviewers’ comments. Furthermore, according to the editor and reviewers’ comments, the relevant regulations had been made in the original manuscript. We also responded point by point to the editor’s and each reviewer’s comments as listed below, along with a clear indication of the revision’s location.

Thank you.

Sincerely yours!

Mingzhen Wang

List of Major Changes (LMC): 

LMC01: The manuscript meets PLOS ONE's style requirements, including those for file naming.

LMC02: The framework levels in the original manuscript are 1 Introduction, 2 Tank model and its Dynamic characteristic, 3 Sloshing wave height of the liquid surface, 4 Seismic ground motion input, 5 Hydrostatic pressure, 6 Hydrodynamic pressure under unidirectional horizontal seismic action, 7 Hydrodynamic pressure under bi-directional horizontal seismic action, 8 Conclusions，Acknowledgments，and References. The framework is confusing, resulting in unclear content levels. The contents of each part are integrated, summarized and modified, and incorporated into the new framework of the revised manuscript. The framework level of the revised manuscript is modified in turn to 1 Introduction, 2 Literature review, 3 Methodology, 4 Results, 5 Discussions, 6 Conclusions, Supporting information, Acknowledgments, Author Contributions, Funding, Competing interests, and References.

The Abstract is emphatically revised. (on Page 1)

The Introduction is added to illustrate the significance of this study. (on Page 1)

LMC03: Full details of the ADINA finite element model are added in Section 3.4 of the revised manuscript. (on Pages 7 and 8)

LMC04: Two pairs of classical ground motions are added as the inputs. The causes of ground motion selection are elucidated, the response spectra are analyzed, and the relevant finite element calculations are completed. (on Pages 9 to 10, on Pages 13 to 22)

LMC05: The necessary references are added. The number of references increased from 15 to 36. (on Pages 24 to 26)

LMC06: Zengshun Chen is added as the second author. And Lin Gao and Zengshun Chen are added as the second unit. (on Page 1)

Response to Editor:

Thanks for your comments on our paper. We have revised our paper according to your comments:

1. There are some grammatical errors in the manuscript. Please recheck the English of the manuscript.

Answer: Thank you very much for your suggestion.

The grammars of the full text are checked by manual check and copyrighted Grammarly software. Grammatical errors include lack of articles, singular or plural predicate errors, complex sentences, and so on.

2. The background and importance of the research should clearly discuss the importance of the research. A little literature is given.

Answer: Thank you very much for your suggestion.

In the original manuscript, the introduction and literature review are mixed, and the introduction section is weakened. Accordingly, on the recommendation of editors and reviewers, the Introduction in the original manuscript is split and supplemented into two sections, the Introduction and the literature review. So the problem that the background and importance of the research are not clearly discussed is solved. The introduction section of the revised manuscript focuses on the background and importance of the research, and the corresponding literatures are supplemented.

The specific modifications are as follows.

1 Introduction

Liquid storage structures widely exist in municipal engineering, petrochemical engineering, and nuclear engineering. Representative liquid storage structures include water storage tanks in water supply and drainage systems, oil storage tanks in the petrochemical industry, and liquid storage tanks in the nuclear industry [1]. These structures are functional structures and play an important role in the industry. However, the liquid storage structure is prone to structural damage and functional damage under previous strong earthquakes. And the resulting indirect loss is far greater than the direct loss [2]. Different from the analysis of the pier in the outer waters, the coupling between the structure and the inner waters belongs to the internal flow problem [3]. Structures and internal liquids exhibit different vibration characteristics when strong earthquakes occur. Liquid inertia and viscosity can dissipate part of the energy and play a certain energy dissipation effect, but at the same time, liquid sloshing will produce hydrodynamic pressure on the structure [4-5]. Different from the ordinary building structure, the existence of liquid greatly improves the natural vibration period of the liquid-structure coupling system. The sloshing mechanism of liquid under long-period ground motion is different from that under short-period ground motion [6-7]. The factors affecting the sloshing characteristics include objective factors such as site, epicentral distance, earthquake magnitude, and earthquake source characteristics, and subjective factors such as structure shape, size, and liquid storage height [8]. There are few studies on the distribution of hydrodynamic pressure for liquid storage structures under long-period ground motion action or bi-directional horizontal ground motion action [6, 9-11].

In order to avoid and reduce the direct, indirect, and secondary disasters caused by the damage of liquid storage structures in an earthquake, the seismic problem of liquid storage structures needs to be paid more attention. It is urgent to further study the liquid sloshing mechanism under the action of bi-directional horizontal ground motion with long period characteristics and establish a feasible and conservative calculation method of hydrodynamic pressure. The research results have important reference values for the safety and economical design of liquid storage structures.

3. Several references such as ACI 350.3-06, API standard 650 (11th Edition), AWWA standard D100-96, (GB 50032-2003)”, and several more are not cited in the text. Please use cite them properly.

Answer: Thank you very much for your suggestion.

The relevant codes were not quoted in the original manuscript. The references are added and cited at the location of code description in the revised manuscript, and the newly added references are as follows.

13. American Concrete Institute (ACI). Seismic design of liquid-containing concrete structures (ACI 350.3-01) and commentary (350.3R-01). American Concrete Institute, Farmington Hills, Mich., 2001.

14. American Petroleum Institute (API). Welded tanks for oil storage (API 650). Washington D.C., America, 2012.

15. AWWA standard D100-96. Welded Steel Tanks for Water Storage. American WaterWorks Association, Denver, Cororado, 1996.

25. Ministry of Housing and Urban-Rural Development of the People’s Republic of China. Code for design of vertical cylindrical welded steel oil tanks (GB 50341-2014) [S]. Beijing, China, 2014. (in Chinese)

31. Ministry of Housing and Urban-Rural Development of the People’s Republic of China. Code for seismic design of outdoor water supply, sewerage, gas and heating engineering (GB 50032-2003). Beijing, China, 2003. (in Chinese)

35. Japanese Industrial Standard. Welded steel tanks for oil storage (JIS B 8501-1985), Japan, 1985.

36. Beijing Municipal Commission of Housing and Urban-Rural Development. Code for seismic design of outdoor water supply, sewerage, gas thermal engineering (TJ32-78). Beijing, China, 1979. (in Chinese)

4. Detail of Finite Element Modeling is missing. Even in reference 13, it seems incomplete.

Answer: Thank you very much for your suggestion.

The detailed finite element model of the tanks is supplemented in Section 3.4 of the revised manuscript, including structure characteristics, material properties, working conditions, and model constructions. Details are as follows.

3.4 Numerical simulation calculation method of hydrodynamic pressure

The types of the analyzed tanks are ground-rested circular reinforced concrete with capacities of 500m3, 200m3, and 2000m3, hereinafter referred to as tank A, B, and C, respectively. Table 3 lists the structure characteristics of the analyzed tanks. Table 4 lists the corresponding relationship between water storage capacity and water storage height. Table 5 lists the material properties of liquid water.

Table 3. Structure characteristics of the circular tank

Shortname Capacity(m3) Bottom thickness(m) Wall thickness(m) Inner radius(m) The maximum water storage height(m) Reinforcement diameter

A 500 0.3 0.25 6.75 3.5 10mm

B 200 0.3 0.25 4.3 3.5 10mm

C 2000 0.3 0.25 13.5 3.5 10mm

Table 4. The corresponding relationship between water storage capacity and water storage height

Water storage capacity No water 10% 20% 30% 40% 50% 60% 70%

Water storage height (m) 0 0.35 0.70 1.05 1.40 1.75 2.10 2.45

Table 5. Material properties of liquid water

Density (Kg/m3) Bulk modulus (Pa) Damping ratio

1000 2.3×109 0.16%

In order to simplify the calculation condition of the tank, the symbol “A-50%” is used to represent that the 500m3 capacity tank has 1.75 meters water storage height. In ADINA software, ADINA Parasolid geometric modeling method is used to establish the tank model. The tank structure is adopted the 3D-Solid element. The concrete material is simulated by Concrete in ADINA, and the reinforcement is set by the Rebar option in the Truss element. The liquid in the tank is adopted the 3D-Fuild element. Thereinto, liner potential-based element is used in static analysis and modal analysis, and potential-based fluid is used in dynamic analysis. The stress-strain curves of the concrete and reinforcement are shown in Fig 5. The bottom of the tank structure is the fixed constraint. In order to make the grid division uniform and improve computational efficiency, the following method is used to divide the grid. The tank body in the direction of tank wall thickness and the bottom plate thickness is divided into three parts, and the tank body in the circumferential direction is divided into 50 parts. Each 0.35 meters along the tank wall height direction is divided into one portion. The radial direction of tanks A, B, and C are divided into 23, 15, and 37 parts, respectively. The circumferential and radial grids of the liquid in the tank are the same as those in the tank body, and the liquid is divided one portion per 0.35 meters in the height direction. Taking tank A as an example, the finite element models of the tank body, reinforcement bar, 30% water storage, and 70% water storage are listed in Fig 6.

a) The concrete b) The reinforcement

Fig 5. The stress-strain curve of the materials in the numerical simulation

a) Tank body b) Reinforcement bar

c) 30% water storage d) 70% water storage

Fig 6. The finite element models of the tank A

5. On what basis earthquakes are selected? Please provide the response spectrum of each earthquake, and the mean response spectrum of the earthquakes.

Answer: Thank you very much for your suggestion.

The three elements of ground motion include spectrum characteristics, effective peak, and duration time. The spectrum characteristic refers to the amplitude and phase characteristics of each harmonic vibration that composes the ground motion. The spectrum shows the intensity distribution of different frequency components, reflecting the dynamic characteristics of ground motion. Different ground motions have different spectral characteristics. The effective peak value reflects the maximum intensity of the ground motion at a certain moment in the earthquake process, which directly reflects the earthquake force and its vibration energy, and the magnitude of earthquake deformation. It is the scale of the influence of the earthquake on the structure. The effective peak value of the original ground motion can be adjusted according to the actual demand. The duration time is the effective duration of the input seismic acceleration time history curve. The measured physical time of the original ground motion is generally tens of seconds. The small peak value at the beginning or the end of the ground motion has little effect on the structure, and the long duration also indirectly reduces the computational efficiency. Therefore, the original ground motion can be intercepted in the seismic response analysis of the structure. The time length of the intercepted section is generally 5 to 10 times the basic natural vibration period of the analyzed structure.

Since this research is to explore the seismic response characteristics of storage structure under long-period ground motion, the obvious difference in spectrum characteristics, including extremely short period, short period, medium-long period, and long period are selected. At the same time, seven natural ground motions in the 2008 Wenchuan Earthquake in China with the peak acceleration of 1m/s2 are used as input for analyses. Considering the accuracy and efficiency of calculation, the duration time of ground motion is taken as 30 seconds. According to the opinions of the review experts, two groups of classical ground motions, including El Centro and Tianjin ground motions, are added as seismic input. The calculated results are consistent with the above seven pairs of natural ground motions. The correctness of calculation and analysis is verified.

In the revised manuscript, the reason for ground motion selection is added first. The response spectrum of each ground motion and the mean response spectrum of ground motion are also added.

The specific additions are described below.

3.5 Seismic ground motion selection and input method

This research is to explore the seismic response characteristics of storage structure under long-period ground motion, so the obvious difference in spectrum characteristics, including extremely short period, short period, medium-long period, and long period are selected. At the same time, seven natural ground motions in the 2008 Wenchuan Earthquake in China with the peak acceleration of 1m/s2 are used as input for analyses [32]. Considering the accuracy and efficiency of calculation, the duration time of ground motion is taken as 30 seconds. The following information of seismic ground motions is collected and listed in Table 6, including name, collecting stations, site condition, epicentral distance, direction, peak acceleration, the moment of peak acceleration, and predominant period. At the same time, two groups of classical ground motions, including El Centro and Tianjin ground motion, are selected as ground motion inputs. The predominant period is a key parameter for the seismic design of important structures. The surface soil layer has a selective amplification effect on seismic waves of different periods, resulting in the waveform of some periods on the seismic record map being particularly many and good, which is called ‘predominant’, so it is called the predominant period of ground motion. The predominant period is the period where the maximum amplitude of soil vibration may occur, which mainly changes with the geotechnical characteristics of the site. When carrying out the seismic design of the structure, the natural vibration period of the structure and the predominant period of different foundations should be considered to ensure that the natural vibration period of the structure is greatly different from the predominant period of the site [33]. The predominant period of ground motion can be obtained through Fourier transform [34]. (on Page 9)

The response spectra of nine main ground motions, the mean response spectrum of all ground motions, and the design response spectrum are plotted in Fig 8. Among them, the design response spectrum is the long-period seismic design spectrum when Tg is equal to 0.55 seconds in Fig 1. The mean response spectrum in Fig 8 is the 95 % guarantee rate. The mean response spectrum obtained from nine ground motions is larger than the design response spectrum, mainly because the number of ground motions is too small. The design response spectrum mainly considers the safety of structure design and takes into account the economic and cost factors. (on Page 10)

Fig 8. Response spectrums, mean response spectrum and design response spectrum of earthquakes

6. What is the mass participation percentage of the modes in Table 1?

Answer: Thank you very much for your suggestion.

Table 1 in the original manuscript is changed to Table 7 in the revised manuscript. The last column in Table 7 is added to represent the mass participation percentage of first-order sloshing period. The modified Table 7 is shown in the following table.

Table 7. Brief tank model features and dynamic characteristics results

Working condition Tank materials Tank inner radius (m) Water storage heights (m) First-order sloshing period (s) Mass participation percentage of the first-order sloshing period (%)

A-10% Reinforced concrete 6.75 0.35 12.201 15.75

A-30% 6.75 1.05 7.128 33.34

A-50% 6.75 1.75 5.650 41.55

A-70% 6.75 2.45 4.929 44.84

B-70% 4.3 2.45 3.368 34.07

C-70% 13.5 2.45 9.461 55.08

7. Gal is not part of an International System of Units (SI). Please change the unit of acceleration to SI.

Answer: Thank you very much for your suggestion.

Gal is commonly used in Earthquake Engineering to describe seismic acceleration, and 1 cm/s2 is 1 gal. When gal is converted to m/s2, 1 gal is 10-2 m/s2. The texts, tables, and figures of gal units in the revised manuscript are changed to International System unit m/s2. Specifically, 100-gal in texts is modified to 1.0 m/s2, and gal in tables and figures is modified to 10-2 m/s2.

Response to Reviewer #1:

Thanks for your comments on our paper. We have revised our paper according to your comments:

1. Add and specify numerical model in ADINA

Answer: Thank you very much for your suggestion.

Complete details on the numerical tank model in ADINA are missing in Section 2 of the original manuscript. Complete details and necessary drawings of the tank models are provided in Section 3.4 of the revised manuscript, so that the readers can understand all the information about the tank models.

The details in Section 3.4 of the revised manuscript are as follows. (on Pages 7 and 8)

3.4 Numerical simulation calculation method of hydrodynamic pressure

The types of the analyzed tanks are ground-rested circular reinforced concrete with capacities of 500m3, 200m3, and 2000m3, hereinafter referred to as tanks A, B, and C, respectively. Table 3 lists the structure characteristics of the analyzed tanks. Table 4 lists the corresponding relationship between water storage capacity and water storage height. Table 5 lists the material properties of liquid water.

Table 3. Structure characteristics of the circular tank

Shortname Capacity(m3) Bottom thickness(m) Wall thickness(m) Inner radius(m) The maximum water storage height(m) Reinforcement diameter

A 500 0.3 0.25 6.75 3.5 10mm

B 200 0.3 0.25 4.3 3.5 10mm

C 2000 0.3 0.25 13.5 3.5 10mm

Table 4. The corresponding relationship between water storage capacity and water storage height

Water storage capacity No water 10% 20% 30% 40% 50% 60% 70%

Water storage height (m) 0 0.35 0.70 1.05 1.40 1.75 2.10 2.45

Table 5. Material properties of liquid water

Density (Kg/m3) Bulk modulus (Pa) Damping ratio

1000 2.3×109 0.16%

In order to simplify the calculation condition of the tank, the symbol “A-50%” is used to represent that the 500m3 capacity tank has 1.75 meters water storage height. In ADINA software, ADINA Parasolid geometric modeling method is used to establish the tank model. The tank structure is adopted the 3D-Solid element. The concrete material is simulated by Concrete in ADINA, and the reinforcement is set by the Rebar option in the Truss element. The liquid in the tank is adopted the 3D-Fuild element. Thereinto, liner potential-based element is used in static analysis and modal analysis, and potential-based fluid is used in dynamic analysis. The stress-strain curves of the concrete and reinforcement are shown in Fig 5. The bottom of the tank structure is the fixed constraint. In order to make the grid division uniform and improve computational efficiency, the following method is used to divide the grid. The tank body in the direction of tank wall thickness and the bottom plate thickness is divided into three parts, and the tank body in the circumferential direction is divided into 50 parts. Each 0.35 meters along the tank wall height direction is divided into one portion. The radial direction of tanks A, B, and C are divided into 23, 15, and 37 parts, respectively. The circumferential and radial grids of the liquid in the tank are the same as those in the tank body, and the liquid is divided one portion per 0.35 meters in the height direction. Taking tank A as an example, the finite element models of the tank body, reinforcement bar, 30% water storage, and 70% water storage are listed in Fig 6.

a) The concrete b) The reinforcement

Fig 5. The stress-strain curve of the materials in the numerical simulation

a) Tank body b) Reinforcement bar

c) 30% water storage d) 70% water storage

Fig 6. The finite element models of the tank A

2. Correct text in chapter 5, Table lists the hydrostatic values, not hydrodynamic values.

Answer: Thank you very much for your suggestion.

Thank you very much for your suggestion. Section 5 of the original manuscript mainly describes the processes and results comparisons of the theoretical calculation and ADINA calculation for hydrostatic pressure. The word “hydrodynamic” in Table 3 of the original manuscript does cause ambiguity. Therefore, the section “Hydrostatic pressure” is modified completely, including text description, figure, and table. “Hydrostatic pressure” is Section 4.2 in the revised manuscript, and as follows. 

4.2 Hydrostatic pressure

In order to verify the correctness of the numerical simulation calculation results, the hydrostatic pressure is calculated by taking the A-50% condition as an example. The calculation formula of hydrostatic pressure is Eq (18).

 (18)

Where, pstatic is hydrostatic pressure. The parameter ρ is the density of the liquid. The density of water is 1000kg/m3. The parameter g is the acceleration of gravity, and its value is 9.81m/s2. The parameter z is the depth of the extraction point.

The locations of extraction points on the tank wall and bottom for the hydrodynamic and hydrostatic pressure are shown in Fig 10. For the working condition of A-50%, the z value at the liquid surface position is 0, so the hydrostatic pressure is 0. The z value of position 1 on the tank wall is 0.35 meters, and the hydrostatic pressure was equal to the multiplication of parameters ρ, g and z, that is, 1000 kg/m3 multiplied by 9.81 m/s2 multiplied by 0.35 meters equal to 3433.5 Pa. The hydrostatic pressure at other extraction points is calculated by this analogy. The calculation results are listed in Column 2 of Table 8. ADINA calculation model can be seen in Section 3.4. Static mode is selected when calculating hydrostatic pressure, and Dynamic-Implicit mode is selected when calculating hydrodynamic pressure. After static calculation in ADINA, the pressure at the corresponding position is extracted as the hydrostatic pressure, and the results are listed in Column 3 of Table 8. For Dynamic-Implicit calculation in ADINA, the pressure at the corresponding position is extracted as the total pressure, the hydrodynamic pressure at a point is equal to the total pressure minus the hydrostatic pressure at this point. Therefore, it is necessary to calculate the hydrostatic pressure before calculating the hydrodynamic pressure at each point.

3. Add hydrodynamic values from numerical model in ADINA

Answer: Thank you very much for your suggestion.

Two pairs of classical ground motions, including El Centro and Tianjin ground motions, are added to calculate the dynamic response of tank models under unidirectional and bi-directional horizontal seismic action. In addition to showing the distributions of hydrodynamic pressure in the figures, the tables of hydrodynamic pressure in ADINA are added in Sections 4.2, 4.3, and S1 File. At the same time, the hydrodynamic pressure distribution maps of all working conditions in Figs 11 to 15 in the original manuscript are supplemented completely. The specific additions are shown below.

The following contents are added to Section 4.3.1.

Therefore, only the maximum values of the hydrodynamic pressures at the tank wall or the radial and circumferential directions of the tank bottom are considered. Taking A-50 % working condition as an example, Table 9, Table 10, and Table 11 respectively list the maximum hydrodynamic pressure of tank wall, tank bottom radial direction, and circumferential direction under unidirectional main seismic motion with 1.0 m/s2 peak acceleration. Due to a large amount of data, the data results of other working conditions are listed in the S1 Table to S15 Table of the S1 File. The maximum distribution maps of the hydrodynamic pressures for all working conditions at 1.0 m/s2 peak acceleration are given in this section in Figs 11-13, respectively. (on Page 13, the last eight lines of the first paragraph in Section 4.3.1)

The following contents are added to Section 4.4.

Table 13 and Table 14 respectively list the maximum hydrodynamic pressure of the whole tank under unidirectional main and secondary seismic motion with 1.0 m/s2 peak acceleration. Table 15 lists the maximum hydrodynamic pressure of the whole tank under bi-directional seismic motions with 1.0 m/s2 peak acceleration.

Table 13. The maximum hydrodynamic pressure PWD,x of the whole tank under unidirectional main seismic motion with 1.0 m/s2 peak acceleration (KPa)

Working condition HSDB-EW JZGYF-NS PJW-NS BJ-EW CC-NS YL-NS HX-NS EL-NS TJ-NS

A-10% 0.757 1.086 1.148 0.853 1.555 1.761 2.412 0.670 0.744

A-30% 1.892 2.391 2.226 1.710 3.028 7.404 3.926 1.986 1.754

A-50% 3.151 2.618 3.752 3.151 4.282 5.615 7.124 3.329 2.805

A-70% 4.417 3.701 4.280 4.099 6.555 8.268 6.856 4.819 3.804

B-70% 4.493 3.930 4.468 4.888 4.368 6.251 6.298 4.914 4.127

C-70% 4.100 4.031 4.284 4.028 4.722 6.670 5.269 4.612 3.667

Table 14. The maximum hydrodynamic pressure PWD,y of the whole tank under unidirectional secondary seismic motion with 1.0 m/s2 peak acceleration (KPa)

Working condition HSDB-NS JZGYF-EW PJW-EW BJ-NS CC-EW YL-EW HX-EW EL-EW TJ-EW

A-10% 0.783 0.906 1.636 0.669 0.884 1.140 0.867 0.905 0.533

A-30% 1.800 1.888 2.553 1.579 1.829 3.269 3.284 2.109 1.103

A-50% 3.079 3.255 4.688 2.868 3.913 5.005 7.190 3.938 1.781

A-70% 3.974 4.385 6.602 3.875 6.175 5.654 6.095 4.296 2.480

B-70% 3.759 3.795 4.527 3.130 4.274 4.908 5.483 4.230 2.420

C-70% 3.800 4.047 5.559 3.224 4.052 5.244 3.741 3.736 2.339

Table 15. The maximum hydrodynamic pressure PWD,2 of the whole tank under bi-directional seismic motions with 1.0 m/s2 peak acceleration (KPa)

Working condition HSDB2 JZGYF2 PJW2 BJ2 CC2 YL2 HX2 EL2 TJ2

A-10% 0.838 1.472 1.945 0.896 1.673 2.348 2.554 1.041 0.802

A-30% 1.939 2.475 2.888 1.711 3.500 8.087 5.010 2.744 1.845

A-50% 3.170 3.220 5.513 3.250 4.450 7.490 8.340 4.329 3.002

A-70% 4.516 4.886 7.778 4.225 7.266 9.470 7.411 5.183 4.084

B-70% 4.768 4.483 5.497 4.921 4.744 7.750 8.015 5.345 4.179

C-70% 4.317 5.460 6.729 4.049 4.956 8.471 6.754 4.936 3.991

The 15 Tables are added to the S1 File. The details can be seen in the S1 File.

4. Correct arrows in Fig. 12

Answer: Thank you very much for your suggestion.

The direction of the arrow is misdrawn in Fig 12a) of the original manuscript. The following left graph is original and error, and the right graph is correct after the arrow is modified. The incorrect graph is replaced in Fig 16a) in Section 4.3 of the revised manuscript.

Incorrect graph Correct graph

5. Why is the non-typical pressure C-70-HX-NS IN Fig. 11b?

Answer: Thank you very much for your suggestion.

The pressure C-70-HX-NS in Fig 11b of the original manuscript is moved to Fig 11g of the revised manuscript. By carefully checking the hydrodynamic pressure data, it is found that the hydrodynamic pressure value 5.269 at a depth of 1.75 meters is mistakenly input to 5.669, resulting in a mutation of the hydrodynamic pressure value in the original figure, as shown in the left figure below. The modified hydrodynamic pressure presents a continuous distribution from the tank bottom to the liquid surface, as shown in the right figure below.

Original picture Revised picture in Fig 11g

The maximum hydrodynamic pressure distributions of the tank wall at different tank radii under HSDB-EW and HX-NS ground motions are listed only in Fig 11 of the original manuscript. All hydrodynamic pressure distributions for the original seven pairs of ground motions and the new two pairs of ground motions are drawn in the revised manuscript as shown below. Through comparative analysis, it can be seen that under the short-period seismic motion action, no matter how large the tank radius is, the hydrodynamic pressure distributed along the tank wall reaches the maximum at the tank bottom position, as shown in Fig 15a), b), c), d), h), i) of the revised manuscript. As the period increases, the hydrodynamic pressure at the liquid surface increases gradually. Under the long-period seismic motion action, the maximum position of hydrodynamic pressure is generally near the liquid surface or near the tank bottom, as shown in Fig 15e), f), g) of the revised manuscript. Under long-period seismic motion action, the influence of tank radius on hydrodynamic pressure of tank wall has not yet shown obvious regularity.

6. Add comparison of individual solutions (numerical, theoretical, and code)

Answer: Thank you very much for your suggestion.

In Section 4.3.2, taking A-50 % working condition as an example, the differences of hydrodynamic pressure in the theoretical calculation, code calculation, and numerical simulation calculation are compared.

4.3.2 Results comparison of the numerical, theoretical, and code methods

Taking A-50 % condition as an example, the hydrodynamic pressures of numerical simulation calculation, theoretical calculation, and code calculation are compared. A comparison of the theoretical and numerical calculation results in Fig 11c) and Fig 12c) show that the hydrodynamic pressures at the tank wall and tank bottom are more similar under short-period seismic motion action. The numerical calculation results of the hydrodynamic pressure at the tank wall exhibit the characteristics of a low liquid level and large tank bottom, and those at the tank bottom show distribution characteristics of an inverse tangent function near the origin. With an increase in the long-period components in the input ground motion, the numerical results become significantly higher than the theoretical results, and the maximum ratio of the two reaches 2.6. The numerical calculation results for the hydrodynamic pressure at the tank wall show the distribution characteristics of a high liquid level and small tank bottom, and those at the tank bottom show the distribution characteristics of a sin function near the origin. The maximum hydrodynamic pressure of the X-axis forward direction for A-50% condition is 0.687 KPa by using the Chinese code GB50032-2003 [31] method in Section 3.3. Comparing the Chinese code GB50032-2003 and the numerical calculation results in Fig 11c) and Fig 13, it can be seen that the code values are obviously small, and the numerical results are almost symmetrical along the vertical to the seismic input direction. The hydrodynamic pressure at the tank bottom is least along the vertical to the seismic input direction. From here, the hydrodynamic pressure at each point gradually increases along an arc and reaches a maximum along with the directions of the ground motion input. The above analyses show that the existing theoretical method for calculating hydrodynamic pressure cannot meet user demand and is unsafe under most working conditions. Therefore, a more accurate method for calculating hydrodynamic pressure must be established.

Response to Reviewer #2:

Thanks for your comments on our paper. We have revised our paper according to your comments:

The study is useful but it seems incomplete. The variation in impulsive and convective pressure under the selected earthquake can not provide a conclusion. The convective component not only depends on liquid depth but also on the dominant frequencies of the earthquake. Hence, adding more earthquake to the investigation is necessary.

Answer: Thank you very much for your suggestion.

Convective hydrodynamic pressure is proportional to the maximum sloshing wave height under seismic ground motion action. The maximum sloshing wave height is related to the spectrum characteristics of ground motion.

The three elements of ground motion include spectrum characteristics, effective peak, and duration time. The spectrum characteristic refers to the amplitude and phase characteristics of each harmonic vibration that composes the ground motion. The spectrum shows the intensity distribution of different frequency components, reflecting the dynamic characteristics of ground motion. Different ground motions have different spectral characteristics. The effective peak value reflects the maximum intensity of the ground motion at a certain moment in the earthquake process, which directly reflects the earthquake force and its vibration energy, and the magnitude of earthquake deformation. It is the scale of the influence of the earthquake on the structure. The effective peak value of the original ground motion can be adjusted according to the actual demand. The duration time is the effective duration of the input seismic acceleration time history curve. The measured physical time of the original ground motion is generally tens of seconds. The small peak value at the beginning or the end of the ground motion has little effect on the structure, and the long duration also indirectly reduces the computational efficiency. Therefore, the original ground motion can be intercepted in the seismic response analysis of the structure. The time length of the intercepted section is generally 5 to 10 times the basic natural vibration period of the analyzed structure.

Since this research is to explore the seismic response characteristics of storage structure under long-period ground motion, the obvious difference in spectrum characteristics, including extremely short period, short period, medium-long period, and long period are selected. At the same time, seven natural ground motions in the 2008 Wenchuan Earthquake in China with the peak acceleration of 1m/s2 are used as input for analyses. Considering the accuracy and efficiency of calculation, the duration time of ground motion is taken as 30 seconds.

In your opinion, two groups of classical ground motion, including El Centro and Tianjin, are added as seismic input. In the revised manuscript, the reason for ground motion selection is added first. The response spectrum of each ground motion and the mean response spectrum of ground motion are also added. The results of hydrodynamic pressure under unidirectional and bi-directional ground motion are obtained, respectively, as shown in Table 9 to Table 11, Fig 11 to Fig 15 in Section 4.3, and Table 13 to Table 15 in Section 4.4 in the revised manuscript. The calculated results are consistent with the above seven pairs of natural ground motions. The correctness of calculation and analysis is verified.

Response to Reviewer #3:

Thanks for your comments on our paper. We have revised our paper according to your comments:

This paper presents a study on circular RC tanks under bi-directional shaking. While the topic is important, it is unclear what the novelty is. Many details regarding analysis and modelling are missing. Specific comments are below.

1. Abstract: Novelty in method, structure considered, and results is unclear.

Answer: Thank you very much for your suggestion.

In the original manuscript, the innovation points of the paper are not prominent, the structure level is chaotic, and the expression of the results is not clear. In the revised manuscript, the Abstract is completely revised, ambiguous statements are removed, and necessary descriptions of innovations and results are added. The revised Abstract is shown below.

Abstract: The research object is the ground-rested circular RC tank. The innovation is to reveal the hydrodynamic pressure law of ground-rested circular RC tanks under bi-directional horizontal seismic action. The relationship between the sloshing wave height and hydrodynamic pressure is determined, the hydrodynamic pressure components and their combination are verified, calculation methods for hydrodynamic pressure are developed, and their distribution laws are presented. The results show that convective hydrodynamic pressure cannot be ignored when the tank is subjected to seismic action. Hydrodynamic pressure under unidirectional horizontal seismic action in X or Y direction is obtained by square root of the sum of impulsive pressure squared and convective pressure squared. Total hydrodynamic pressure under bi-directional horizontal seismic action is obtained by the square root of the sum of X-direction hydrodynamic pressure squared and Y-direction hydrodynamic pressure squared. This method can ensure the accuracy and reliability of hydrodynamic pressure calculation.

2. Introduction: The gap in the state-of-the-art and specific questions addressed are not clearly identified in the Introduction section.

Answer: Thank you very much for your suggestion. 

The framework level in the original manuscript is chaotic, which leads to the unclear content level of the paper. According to your above suggestions, the Introduction Section in the original manuscript is modified into the Literature Review Section in the revised manuscript. The Introduction Section is added in the revised manuscript, including the research background, research object, research status (namely the gap in the state-of-the-art and specific questions), research significance, and research purpose. The Introduction Section of the revised manuscript is as follows.

1 Introduction

Liquid storage structures widely exist in municipal engineering, petrochemical engineering, and nuclear engineering. Representative liquid storage structures include water storage tanks in water supply and drainage systems, oil storage tanks in the petrochemical industry, and liquid storage tanks in the nuclear industry [1]. These structures are functional structures and play an important role in the industry. However, the liquid storage structure is prone to structural damage and functional damage under previous strong earthquakes. And the resulting indirect loss is far greater than the direct loss [2]. Different from the analysis of the pier in the outer waters, the coupling between the structure and the inner waters belongs to the internal flow problem [3]. Structures and internal liquids exhibit different vibration characteristics when strong earthquakes occur. Liquid inertia and viscosity can dissipate part of the energy and play a certain energy dissipation effect, but at the same time, liquid sloshing will produce hydrodynamic pressure on the structure [4-5]. Different from the ordinary building structure, the existence of liquid greatly improves the natural vibration period of the liquid-structure coupling system. The sloshing mechanism of liquid under long-period ground motion is different from that under short-period ground motion [6-7]. The factors affecting the sloshing characteristics include objective factors such as site, epicentral distance, earthquake magnitude, and earthquake source characteristics, and subjective factors such as structure shape, size, and liquid storage height [8]. There are few studies on the distribution of hydrodynamic pressure for liquid storage structures under long-period ground motion action or bi-directional horizontal ground motion action [6, 9-11].

In order to avoid and reduce the direct, indirect, and secondary disasters caused by the damage of liquid storage structures in an earthquake, the seismic problem of liquid storage structures needs to be paid more attention. It is urgent to further study the liquid sloshing mechanism under the action of bi-directional horizontal ground motion with long period characteristics and establish a feasible and conservative calculation method of hydrodynamic pressure. The research results have important reference values for the safety and economical design of liquid storage structures.

3. Table 1: Should it be “Tank model storage” instead of “Tank model shortage”?

Answer: Thank you very much for your suggestion.

The use of “Tank model shortage” as the column name may cause ambiguity. The first column in Table 1 indicates the distinction of working conditions. Therefore, the first column name in Table 1 is replaced with “Working condition” in the revised manuscript. “Table 1” in the original manuscript becomes Table 7 in Section 4.1 of the revised manuscript.

4. Section 2: Complete details on the geometric and material properties of the tank should be provided, so that the results can be simulated by others. Necessary drawings should also be provided.

Answer: Thank you very much for your suggestion. 

Complete details on the geometric and material properties of the tank are missing in Section 2 of the original manuscript. Complete details and necessary drawings of the tanks are provided in Section 3.4 of the revised manuscript, so that the results can be simulated by readers.

The details in Section 3.4 of the revised manuscript are as follows.

3.4 Numerical simulation calculation method of hydrodynamic pressure

The types of the analyzed tanks are ground-rested circular reinforced concrete with capacities of 500m3, 200m3, and 2000m3, hereinafter referred to as tanks A, B, and C, respectively. Table 3 lists the structure characteristics of the analyzed tanks. Table 4 lists the corresponding relationship between water storage capacity and water storage height. Table 5 lists the material properties of liquid water.

Table 3. Structure characteristics of the circular tank

Shortname Capacity(m3) Bottom thickness(m) Wall thickness(m) Inner radius(m) The maximum water storage height(m) Reinforcement diameter

A 500 0.3 0.25 6.75 3.5 10mm

B 200 0.3 0.25 4.3 3.5 10mm

C 2000 0.3 0.25 13.5 3.5 10mm

Table 4. The corresponding relationship between water storage capacity and water storage height

Water storage capacity No water 10% 20% 30% 40% 50% 60% 70%

Water storage height (m) 0 0.35 0.70 1.05 1.40 1.75 2.10 2.45

Table 5. Material properties of liquid water

Density (Kg/m3) Bulk modulus (Pa) Damping ratio

1000 2.3×109 0.16%

In order to simplify the calculation condition of the tank, the symbol “A-50%” is used to represent that the 500m3 capacity tank has 1.75 meters water storage height. In ADINA software, ADINA Parasolid geometric modeling method is used to establish the tank model. The tank structure is adopted the 3D-Solid element. The concrete material is simulated by Concrete in ADINA, and the reinforcement is set by the Rebar option in the Truss element. The liquid in the tank is adopted the 3D-Fuild element. Thereinto, liner potential-based element is used in static analysis and modal analysis, and potential-based fluid is used in dynamic analysis. The stress-strain curves of the concrete and reinforcement are shown in Fig 5. The bottom of the tank structure is the fixed constraint. In order to make the grid division uniform and improve computational efficiency, the following method is used to divide the grid. The tank body in the direction of tank wall thickness and the bottom plate thickness is divided into three parts, and the tank body in the circumferential direction is divided into 50 parts. Each 0.35 meters along the tank wall height direction is divided into one portion. The radial direction of tanks A, B, and C are divided into 23, 15, and 37 parts, respectively. The circumferential and radial grids of the liquid in the tank are the same as those in the tank body, and the liquid is divided one portion per 0.35 meters in the height direction. Taking tank A as an example, the finite element models of the tank body, reinforcement bar, 30% water storage, and 70% water storage are listed in Fig 6.

a) The concrete b) The reinforcement

Fig 5. The stress-strain curve of the materials in the numerical simulation

a) Tank body b) Reinforcement bar

c) 30% water storage d) 70% water storage

Fig 6. The finite element models of the tank A

5. Fig. 2: Response spectra of ground motions should be compared with target response spectrum.

Answer: Thank you very much for your suggestion. 

Response spectra of ground motions are added and compared. The details are as follows.

The response spectra of nine main ground motions, the mean response spectrum of all ground motions, and the design response spectrum are plotted in Fig 8. Among them, the design response spectrum is the long-period seismic design spectrum when Tg is equal to 0.55 seconds in Fig 1. The mean response spectrum in Fig 8 is the 95 % guarantee rate. The mean response spectrum obtained from nine ground motions is larger than the design response spectrum, mainly because the number of ground motions is too small. The design response spectrum mainly considers the safety of structure design and takes into account the economic and cost factors. (The second paragraph in Section 3.5)

Fig 8. Response spectrums, mean response spectrum and design response spectrum of earthquakes

6. Section 4: What is the definition of predominant period?

Answer: Thank you very much for your suggestion.

The predominant period is a key parameter for the seismic design of important structures. The surface soil layer has a selective amplification effect on seismic waves of different periods, resulting in the waveform of some periods on the seismic record map being particularly many and good, which is called ‘predominant’, so it is called the predominant period of ground motion. The predominant period is the period where the maximum amplitude of soil vibration may occur, which mainly changes with the geotechnical characteristics of the site. When carrying out the seismic design of the structure, the natural vibration period of the structure and the predominant period of different foundations should be considered to ensure that the natural vibration period of the structure is greatly different from the predominant period of the site. The predominant period of ground motion can be obtained through Fourier transform. 

And the above definition of the predominant period is added to line 9 of Section 3.5 in the revised manuscript.

7. Table 3 presents the results obtained using theoretical expressions and ADINA software. However, no detail on the theoretical expressions and software modelling are provided so far.

Answer: Thank you very much for your suggestion. 

In Section 3.4 of the revised manuscript, all the information of the tank ADINA model is added, and the ADINA calculation results in Table 3 in the original manuscript depend on this model. The calculation formula of hydrostatic pressure is not given in the original manuscript. Therefore, in Section 4.2 of the revised manuscript, the calculation formula of hydrostatic pressure and the meaning of each parameter is supplemented. It also supplements the calculation process and results of ADINA model. Due to the modification of the whole paper framework, Section 5 in the original manuscript becomes Section 4.2 in the revised manuscript, and Table 3 in the original manuscript becomes Table 8 in the revised manuscript. The main contents of section 4.2 are revised as follows.

In order to verify the correctness of the numerical simulation calculation results, the hydrostatic pressure is calculated by taking the A-50% condition as an example. The calculation formula of hydrostatic pressure is Eq (18).

 (18)

Where, pstatic is hydrostatic pressure. The parameter ρ is the density of the liquid. The density of water is 1000kg/m3. The parameter g is the acceleration of gravity, and its value is 9.81m/s2. The parameter z is the depth of the extraction point.

The locations of extraction points on the tank wall and bottom for the hydrodynamic and hydrostatic pressure are shown in Fig 10. For the working condition of A-50%, the z value at the liquid surface position is 0, so the hydrostatic pressure is 0. The z value of position 1 on the tank wall is 0.35 meters, and the hydrostatic pressure was equal to the multiplication of parameters ρ, g and z, that is, 1000 kg/m3 multiplied by 9.81 m/s2 multiplied by 0.35 meters equal to 3433.5 Pa. The hydrostatic pressure at other extraction points is calculated by this analogy. The calculation results are listed in Column 2 of Table 8. ADINA calculation model can be seen in Section 3.4. Static mode is selected when calculating hydrostatic pressure, and Dynamic-Implicit mode is selected when calculating hydrodynamic pressure. After static calculation in ADINA, the pressure at the corresponding position is extracted as the hydrostatic pressure, and the results are listed in Column 3 of Table 8. For Dynamic-Implicit calculation in ADINA, the pressure at the corresponding position is extracted as the total pressure, the hydrodynamic pressure at a point is equal to the total pressure minus the hydrostatic pressure at this point. Therefore, it is necessary to calculate the hydrostatic pressure before calculating the hydrodynamic pressure at each point.

8. Eq. 2, 3: All parameters should be defined.

Answer: Thank you very much for your suggestion. 

The definitions of parameters are checked one by one from the beginning to the end of the revised manuscript. Parameters not defined in the original manuscript are defined in the revised manuscript. The definitions of parameters are added as follows.

βmax in Fig 1 is equal to 2.25. Tg in Fig 1 is site characteristic period, and its values are detailed in Table 1. (The second paragraph in Section 3.1)

Table 1 The values of site characteristic period Tg 

Seismic Design Group Site Classification

 Ⅰ0 Ⅰ1 Ⅱ Ⅲ Ⅳ

First group 0.20 0.25 0.35 0.45 0.65

Second Group 0.25 0.30 0.40 0.55 0.75

Third Group 0.30 0.35 0.45 0.65 0.90

Where: the parameter r is the vertical distance from the calculation point to the centerline of the liquid storage structure. θ is the angle in the circumferential direction. z is the depth of the liquid. t is the time. is velocity potential. And g is the acceleration of gravity. The meaning of the above parameters can be understood by combining Fig 2. (The second paragraph in Section 3.2)

Fig 4. schematic diagram of geometry and coordinate system of the circular tank

n is the unit vector in the normal direction. (The ninth paragraph in Section 3.2)

The parameter ρ is the density of the liquid. a is the inner radius of the liquid storage structure. θ is the angle in the circumferential direction. (The nineteenth paragraph in Section 3.2)

pstatic is hydrostatic pressure. The parameter ρ is the density of the liquid. The density of water is 1000kg/m3. The parameter g is the acceleration of gravity, and its value is 9.81m/s2. The parameter z is the depth of the extraction point. (The second paragraph in Section 4.2)

The definition of c and �W are the same as in Eq (20). (The fifth paragraph in Section 4.3.5)

The definitions of k and �W can be seen the Eq (1) and Eq (20), respectively. (The tenth paragraph in Section 4.3.5)

9. Section 6.1: It is unclear if an existing solution is used, or is it for a new structure and support condition, or if a new approach for formulation is used.

Answer: Thank you very much for your suggestion.

Section 6.1 in the original manuscript has been adjusted to Section 3.2 in the revised manuscript. This section gives the derivation process of the theoretical calculation formula of hydrodynamic pressure, and only describes the theoretical method, which can be used by readers. The results of the hydrodynamic pressure for the tank model calculated by the theoretical method are listed in Fig 9 and Fig 10 in Section 4.3 of the manuscript, which is compared with the numerical simulation results.

Response to Reviewer #4:

Thanks for your comments on our paper. We have revised our paper according to your comments:

The paper attempts to determine the link between sloshing wave height and hydrodynamic pressure, as well as to verify the hydrodynamic pressure components and their combination. The paper developed a method for calculating hydrodynamic pressure and presented its distribution rules. I appreciate the author's efforts, and I have some comments that I believe may help to improve the current form of the article.

1. The paper's structure is unclear. The reader will have difficulty understanding the article in the present format; for example, the introduction takes the form of a literature review.

1-1) Please ensure that you have an introduction section that meets the specifications listed below:

-Indicate the work's field, why it is vital, and what has already been done with proper citations.

-Indicate a gap, raise a research question, or criticise previous work in this area.

-Outline the goal and explain the current study, emphasising what is unique and why it is significant.

Answer: Thank you very much for your in-depth study and analysis of this paper. The amendments you put forward are the important basis for me to modify the whole paper framework. The framework level in the original manuscript is chaotic, which leads to the unclear content level of the paper. According to your above suggestions, the contents of each part are integrated, summarized and modified, and incorporated into the new framework of the revision. The framework level of the revised manuscript is modified as follows, which is 1 Introduction, 2 Literature review, 3 Methodology, 4 Results, 5 Discussions, 6 Conclusions, Supporting information, Acknowledgments, Author Contributions, Funding, Competing interests, and References.

The Introduction Section is modified according to the above three suggestions, as shown in the following two paragraphs. At the same time, ten references are added in this section in addition to a reference in the original manuscript.

1 Introduction

Liquid storage structures widely exist in municipal engineering, petrochemical engineering, and nuclear engineering. Representative liquid storage structures include water storage tanks in water supply and drainage systems, oil storage tanks in the petrochemical industry, and liquid storage tanks in the nuclear industry [1]. These structures are functional structures and play an important role in the industry. However, the liquid storage structure is prone to structural damage and functional damage under previous strong earthquakes. And the resulting indirect loss is far greater than the direct loss [2]. Different from the analysis of the pier in the outer waters, the coupling between the structure and the inner waters belongs to the internal flow problem [3]. Structures and internal liquids exhibit different vibration characteristics when strong earthquakes occur. Liquid inertia and viscosity can dissipate part of the energy and play a certain energy dissipation effect, but at the same time, liquid sloshing will produce hydrodynamic pressure on the structure [4-5]. Different from the ordinary building structure, the existence of liquid greatly improves the natural vibration period of the liquid-structure coupling system. The sloshing mechanism of liquid under long-period ground motion is different from that under short-period ground motion [6-7]. The factors affecting the sloshing characteristics include objective factors such as site, epicentral distance, earthquake magnitude, and earthquake source characteristics, and subjective factors such as structure shape, size, and liquid storage height [8]. There are few studies on the distribution of hydrodynamic pressure for liquid storage structures under long-period ground motion action or bi-directional horizontal ground motion action [6, 9-11].

In order to avoid and reduce the direct, indirect, and secondary disasters caused by the damage of liquid storage structures in an earthquake, the seismic problem of liquid storage structures needs to be paid more attention. It is urgent to further study the liquid sloshing mechanism under the action of bi-directional horizontal ground motion with long period characteristics and establish a feasible and conservative calculation method of hydrodynamic pressure. The research results have important reference values for the safety and economical design of liquid storage structures.

1-2) for the literature review part, you can modify the present introduction to perform as a literature review.

Answer: Thank you very much for your suggestion. 

The introduction section in the original manuscript becomes the literature review section in the revised manuscript. The part contents in Literature Review Section are modified. The first sentence in the first paragraph is deleted. The deleted third paragraph is a discussion on the composition of hydrodynamic pressure, which was integrated into the discussion section of the revised manuscript. The deleted fourth paragraph was integrated into the introduction section of the revised manuscript. In addition, the specifications of different countries mentioned in this section are all cited as references.

1-3) You need a methodology part. The goal is to provide enough information so that a competent researcher can replicate the study. Many of your readers will skip this part since they already know the broad methods you employed from the Introduction. However, careful drafting of this part is required since your results must be repeatable in order to be scientifically valid. Otherwise, your paper is not scientifically sound.

Answer: Thank you very much for your suggestion. 

The methodological section is added to the revised manuscript so as to improve readability and usability, and enhance the scientificity of the paper. The methodology section includes five parts, namely, the calculation method of wave height for liquid sloshing (Section 3 in the original manuscript), the theoretical calculation method of hydrodynamic pressure ( the content related to the method in the original manuscript in Section 6.1 ), the code calculation method of hydrodynamic pressure ( the content related to the method in the original manuscript in Section 6.2 ), the numerical simulation calculation method of hydrodynamic pressure (Section 2 is supplemented and improved in the original manuscript ), and the seismic ground motion selection and input method ( the Section 4 in the original manuscript ). The methodology section focuses on the method, and the solution results of the problems are listed in the result section.

2. The result indicates that the paper's findings are a continuation of previous research such as Mingzhen, W., & Lin, G. (2018) [Dynamic Time-History analyses of Ground Reinforced Concrete Tank in Water Supply System under Bi-directional Horizontal Seismic Actions]. Table 2 and several other data from this study were used without citation. Please Make sure that you maintain the novelty and refrain from repeating the same outcome.

Answer: Thank you very much for your careful search. This study is indeed a continuation of the paper mentioned above. Actually, we are Mingzhen Wang and Lin Gao. In the past nine years, we have been engaged in the study of the seismic dynamic response of liquid storage structures. The research methods include structure tests, theoretical analysis, and numerical simulation.

The ground motion data sources in Table 2 of the original manuscript were not cited. The references, including data sources, are added in Section 3.5 of the revised manuscript. The information of the reference is as follows.

32. Department of Earthquake Disaster Prevention, China Earthquake Administration. Uncorrected acceleration records of Wenchuan 8.0 earthquake [J]. Report on strong earthquake motion records in China, 12(1): 1-302. (in Chinese)

3. Please cite the primary sources of the formulas you utilised in this study.

Answer: Thank you very much for your suggestion. 

There are 24 equations in this paper. Eq (1) came from the Reference [27] is cited in line 2 of paragraph 1 in Section 3.1 of the revised manuscript. Eqs (2) to (16) are quoted from References [29] and [30], both of which are Chinese books, and annotated in the first sentence of paragraph 1 in Section 3.2. Eq (17) is cited from the Chinese code (GB50032-2003), that is the Reference [31], marked in the third line of paragraph 1 in Section 3.3. Eq (18) is the calculation formula of liquid pressure in physics. Eqs (19) to (24) are derived from the calculation results.

---

## [Decision Letter · Decision Letter 1]

4 Mar 2022

Hydrodynamic pressure law of ground-rested circular RC tank under bi-directional horizontal seismic action

PONE-D-21-24114R1

Dear Dr. Wang,

We’re pleased to inform you that your manuscript has been judged scientifically suitable for publication and will be formally accepted for publication once it meets all outstanding technical requirements.

Kind regards,

Ahad Javanmardi, Ph.D

Academic Editor

PLOS ONE

Additional Editor Comments (optional):

Reviewers' comments:

Reviewer's Responses to Questions

**Comments to the Author**

1. If the authors have adequately addressed your comments raised in a previous round of review and you feel that this manuscript is now acceptable for publication, you may indicate that here to bypass the “Comments to the Author” section, enter your conflict of interest statement in the “Confidential to Editor” section, and submit your "Accept" recommendation.

Reviewer #2: All comments have been addressed

Reviewer #3: All comments have been addressed

Reviewer #4: All comments have been addressed

2. Is the manuscript technically sound, and do the data support the conclusions?

Reviewer #2: Partly

Reviewer #3: Yes

Reviewer #4: Yes

3. Has the statistical analysis been performed appropriately and rigorously? 

Reviewer #2: Yes

Reviewer #3: Yes

Reviewer #4: Yes

4. Have the authors made all data underlying the findings in their manuscript fully available?

Reviewer #2: Yes

Reviewer #3: Yes

Reviewer #4: Yes

5. Is the manuscript presented in an intelligible fashion and written in standard English?

Reviewer #2: Yes

Reviewer #3: Yes

Reviewer #4: Yes

6. Review Comments to the Author

Reviewer #2: The authors have addressed all the queries raised by me.So, this manuscript may be accepted for possible publication.

Reviewer #3: (No Response)

Reviewer #4: Thank you for addressing all the comments.

In my opinion, the paper is in the proper format for publication.

Best wishes for your future endeavors.

Stay Safe!

7. PLOS authors have the option to publish the peer review history of their article (what does this mean?). If published, this will include your full peer review and any attached files.

Reviewer #2: No

Reviewer #3: No

Reviewer #4: No

---

## [Editor Report · Acceptance letter]

11 Mar 2022

PONE-D-21-24114R1 

Hydrodynamic pressure law of ground-rested circular RC tank under bi-directional horizontal seismic action 

Dear Dr. Wang:

I'm pleased to inform you that your manuscript has been deemed suitable for publication in PLOS ONE. Congratulations! Your manuscript is now with our production department. 

Kind regards, 

on behalf of

Dr. Ahad Javanmardi 

Academic Editor

PLOS ONE